# A joint proteomic and genomic investigation provides insights into the mechanism of calcification in coccolithophores

Alastair Skeffington[2,3], Axel Fischer[2], Sanja Sviben [2], Magdalena Brzezinka[2], Michał Górka[2], Luca Bertinetti[4], Christian Woehle[5], Bruno Huettel [5], Alexander Graf[2] & André Scheffel [1,2] ✉

Coccolithophores are globally abundant, calcifying microalgae that have profound effects on marine biogeochemical cycles, the climate, and life in the oceans. They are characterized by a cell wall of $CaCO_3$ scales called coccoliths, which may contribute to their ecological success. The intricate morphologies of coccoliths are of interest for biomimetic materials synthesis. Despite the global impact of coccolithophore calcification, we know little about the molecular machinery underpinning coccolithophore biology. Working on the model *Emiliania huxleyi*, a globally distributed bloom-former, we deploy a range of proteomic strategies to identify coccolithogenesis-related proteins. These analyses are supported by a new genome, with gene models derived from long-read transcriptome sequencing, which revealed many novel proteins specific to the calcifying haptophytes. Our experiments provide insights into proteins involved in various aspects of coccolithogenesis. Our improved genome, complemented with transcriptomic and proteomic data, constitutes a new resource for investigating fundamental aspects of coccolithophore biology.

Coccolithophores are globally distributed, calcifying marine microalgae which make substantial contributions to primary productivity and affect global climate as a major component of the carbonate counter pump[1] and producers of DMSP[2]. Owing to their importance as bloom-forming and globally abundant algae, they have become model species for the study of a variety of processes, including algal interactions with bacteria[3] and viruses[4,5], the consequences of infection for carbon export[6], the effects of ocean acidification on algal fitness[7] to the role of life-cycle characteristics in niche differentiation[8], and the dynamics of environmental conditions in the oceans since the mid-Mesozoic[9–11].

The calcite produced by coccolithophores takes the form of scales, with intricate, species-specific morphologies, which are produced inside the cells in a specialized compartment known as the coccolith vesicle (CV)[12]. Calcite crystals nucleate on the rim of an organic base plate within the CV[13,14], and polyanionic polysaccharides that become associated with the calcite during its formation (coccolith-associated polysaccharides, CAPs)[15,16] likely play roles in the delivery of the calcium to the site of crystal growth[17], and may even affect the crystal morphology[18,19]. The pathway of calcium delivery to the CV is uncertain, but likely routes via the endomembrane system to preserve cellular calcium signaling. Recently discovered acidocalcisome-like calcium storage organelles may also play a role in calcium delivery[20,21]. Completed coccoliths are exocytosed onto the cell surface, forming a layer around the cell. The extraordinary morphological patterning of coccoliths at the micro- and the nano-scales means they have potential as components of novel materials[22], and if we can understand the

[1]Technische Universität Dresden, Faculty of Biology, 01307 Dresden, Germany. [2]Max-Planck Institute of Molecular Plant Physiology, Potsdam-Golm 14476, Germany. [3]Biological and Environmental Sciences, University of Stirling, Stirling FK9 4LA, UK. [4]Max Planck Institute of Colloids and Interfaces, Potsdam-Golm 14476, Germany. [5]Max Planck Institute for Plant Breeding Research, Max Planck-Genome-Centre Cologne, Cologne 50829, Germany. ✉e-mail: andre.scheffel@tu-dresden.de

complex control of crystal morphologies that takes place in cocco-lithophores, it may be possible to use this knowledge for developing new methods for producing complex arrays of inorganic crystals for applications in sensing, catalysis, and photonics[23,24].

*Emiliania huxleyi* has emerged as a key model for coccolithophore biology, due to its numerical dominance in coccolithophore communities in the modern ocean[25] and ease of culture in the laboratory. It presents a haplodiplontic life cycle[26], and only calcifies in the diploid phase; the haploid phase is naked of calcite and flagellated. It is still the only coccolithophore with a publicly available sequenced genome[27].

Proteins have been demonstrated to be critical components of the biomineralization machinery in many organisms[28–30]. Two main approaches have been taken to identify coccolithophore calcification proteins to date. First, biochemical analysis of cells and intracellular fractions, which has allowed for the identification of an acidic protein called GPA[31], whose role in calcification is still unknown, as well as a vacuolar-type ATPase proton pump[32] and a voltage-gated $H^+$ channel[33], with putative roles in pH control during calcification. Second, gene-expression-based comparisons of cultures that vary in their calcification state and analysis of the *E. huxleyi* genome, have led to the identification of several candidate proteins potentially involved in calcification[27,34–36], such as CAX transporters and anion exchangers for $Ca^{2+}$ and $HCO_3^-$ uptake respectively. It is clear from the intricate and species-specific morphologies of coccoliths and the complex cell biology of their synthesis that there must be many, as yet undiscovered, proteins involved in coccolith formation, for example, those involved in CV biogenesis and morphogenesis. Although the transcriptomic approaches described above provide clues, they have several limitations in that they do not provide evidence about the existence or abundance of the encoded proteins or their site of action in the cell, nor whether they host posttranslational modifications that might play functional roles.

Here we set out to identify proteins potentially involved in calcification in *E. huxleyi* with a suit of proteomics experiments, allowing high-confidence identifications based on behaviors in orthogonal datasets. The fact that many biomineral associated proteins have low sequence complexity, with biased composition and repetitive elements[37–39], means that their genes also tend to have unusual sequence properties. These pose a challenge to genome annotation and can lead to incorrect or missing gene models, and limit the discovery space of studies relying on that data. To overcome this issue, we underpin our proteomics by annotating a new *E. huxleyi* genome with a multi-condition long-read transcriptome. This allows us to identify many candidate calcification genes that are absent from the original *E. huxleyi* genome, including many that lack known functions.

## Results

### An improved genome and transcriptome improve the identification of coccolithogenesis-related proteins

We wanted to improve on the existing *E. huxleyi* protein predictions (Emihu1[27]) to alleviate the inaccuracies in the gene models noted by others[27,40,41] and to ensure key mineralization proteins were properly represented in our database. We sequenced the genome of *E. huxleyi* using PacBio technology resulting in a diploid genome (Emihu2) of 196 Mb on 600 scaffolds, with 38x average coverage (Table 1, Supplementary Table 1). The genome was sufficiently heterozygous for phasing, resulting in a haploid genome of 98 Mb on 165 scaffolds. This is significantly more contiguous than the current publicly available diploid Emihu1 genome which is on 7809 scaffolds[27].

To annotate the genome and predict the protein sequences encoded, we used PacBio Isoseq-sequencing of cDNA derived from four cultivation conditions (Supplementary Fig. 1), to cover much of the expression potential of the diploid life-cycle stage of the alga. Mapping the Isoseq data from all conditions to the genome yielded models for 44,718 genes on the diploid genome and 22,363 on the haploid. For all genes derived from Isoseq data, we gave names starting

**Table 1 | Key summary statistics of Emihu2 compared to Emihu1**

| | Emihu1 | Emihu2 diploid | Emihu2 haploid |
|---|---|---|---|
| Genome size (Mbp) | 168 | 198 | 98 |
| No. of scaffolds | 7809 | 600 | 165 |
| N50 (bp) | 404,808 | 682,851 | 1,120,510 |
| % GC | 61 | 66,4 | 66.4 |
| % Repeats | 34.05 | 35.82 | - |

with "EhG" for '*E. huxleyi* gene'. There was evidence for multiple iso-forms in just under 40% of the genes (Fig. 1a). More than a third of the identified transcripts were novel, in that they lacked any overlap with Emihu1 gene models when mapped to either the Emihu2 or the Emihu1 genome (Fig. 1b). The low proportion of exact matches between the two transcript sets agrees with previous analyses that the Emihu1 models are often wrong in the details of intron-exon boundaries[41]. Although a saturation analysis (Supplementary Fig. 2) indicated we had identified 85–88% and 84–89% of expressed genes and isoforms respectively, we knew we would still be missing some of the expression potential of the organism, particularly because we were unable to generate a haploid (non-calcifying) clone[34] of this strain for sequencing. To rectify this, we also performed de novo gene prediction using the BRAKER pipeline, trained using our Isoseq data and added those de novo gene models (names start with Br) that did not substantially overlap with Isoseq-derived models to our transcriptome. We used both the Isoseq and genomic sequences for each gene model to predict open reading frames (ORFs) using several tools and selected one ORF per transcript based on completeness, Pfam domain content and length, with extensive manual curation. A substantial proportion of the resulting predicted proteome was novel with respect to the Emihu1 proteome, with over 40% of our proteins lacking a blast hit in the Emihu1 proteome with conservative parameters (Fig. 1c). A BUSCO analysis[42] also suggested our new proteome was more complete (Supplementary Fig. 3, Supplementary Table 2) than the Emihu1 proteome, although the reliability of completeness estimates on under-studied clades such as the haptophytes is questionable.

To assess the performance of the new protein database in a proteomics context, we compared spectral identification rates using the Emihu2 proteome and the publicly available Emihu1 proteome with samples derived from either whole-cell extracts or purified coccoliths. To make between-sample comparisons meaningful, we first removed low-quality spectra using a de novo sequencing quality score cutoff and then clustered the spectra to reduce the over-representation of more abundant peptides. Overall identification rates were only slightly lower than rates achieved using data from the well-studied model organism *Arabidopsis* (Supplementary Fig. 4). We found a small but consistent increase in identification rates from whole-cell extracts using Emihu2 compared to Emihu1 (Fig. 1d). These samples are dominated by abundant proteins involved in photosynthesis and central metabolism, which tend to be well predicted by de novo annotation pipelines. In contrast, given that biomineralization proteins are also often biased in amino acid composition, and low in sequence complexity, we hypo-thesized that they would be less well represented in a largely de novo predicted database. In support of this idea, we found a much greater discrepancy in spectral identification rates between the two databases for the coccolith samples, with a far higher proportion of spectra being identified with the new Emihu2 database (Fig. 1d).

### Phylostratigraphic analysis reveals Calcihaptophycidae-specific genes

To generate a powerful tool to prioritize candidate proteins from our proteomics experiments we performed a Phylostratigraphic analysis

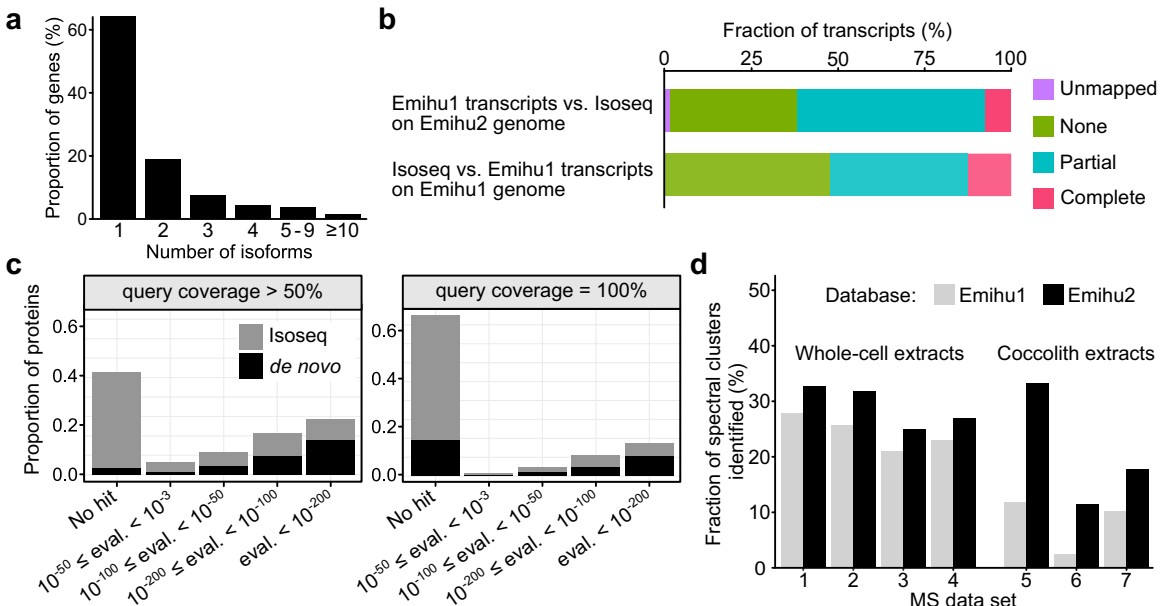

**Fig. 1 | Properties of the *E. huxleyi* genome and proteome v2. a** Numbers of isoforms detected in genes derived from the Isoseq data. **b** Comparison of Emihu1 predicted best gene models and the gene models derived from the Isoseq data using GFFCompare. Emihu1 transcripts were either compared to Isoseq data through mapping to the Emihu2 genome or Isoseq transcripts were compared to Emihu1 transcripts through mapping to the Emihu1 genome. **c** Comparison of the new predicted proteome (from Isoseq derived, plus de novo predicted gene models) to the Emihu1 best predicted proteome using DIAMOND blastp. Proteins were categorized according to the e-value (eval.) of the top High-scoring Segment Pair, requiring a query coverage of either 100% or >50%. **d** Comparison of identification rates of spectra clusters (where only high-quality spectra were used, and each cluster likely represents a unique peptide) using the new *E. huxleyi* protein database Emihu2, and the Emihu1 best proteins database. Rates are shown for four independent *E. huxleyi* whole-cell detergent extracts (1–4) and for three independent extracts derived from isolated coccoliths (5–7).

for the identification of genes specific to the calcifying haptophytes using predicted proteomes from 38 Eukaryotes, focused on the algae and protists, including the Emihu2 proteome (Supplementary Table 3, Supplementary Fig. 5). This allowed us to assign *E. huxleyi* proteins to orthogroups, and to use the distribution of the orthogroups among these extant species to infer the point in the phylogeny at which the orthogroups likely arose (Fig. 2). Many orthogroups were *E. huxleyi* specific (4941, Fig. 2), which may to some extent reflect the fact that similarly detailed genomic and transcriptomic information is not available for other haptophytes. However, it likely also reflects genuine innovation in the *E. huxleyi* clade, as we would expect orthogroups widespread among the haptophytes would have been detected at least once in the transcriptome and genome data from multiple species used in the phylostratigraphic analysis. These recently evolved genes represent 13% of *E. huxleyi* genes, although 46% of those contain known domains, implying more ancient DNA sequence was often involved in their origins, perhaps through processes such as domain fusion. The 724 orthogroups specific to the Calcihaptophycidae clade, which contains all coccolithophores, are of particular interest for understanding the mechanism of calcification.

Given the prevalence of low complexity and disordered protein sequences in biomineralization systems, it was intriguing that sequences in orthogroups arising in the Isochrysidales and the *Gephyrocapsa* clade tend to be unusually low in sequence complexity and, have a strong tendency to disorder. Intriguingly, 2156 orthogroups of haptophyte age or younger have been lost in non-calcifying *Isochrysis*, but are retained in *E. huxleyi*, and 254 of these are of Calcihaptophycidae age. We assessed which Pfam domains were over-represented in each orthogroups-age class relative to the entirety of the *E. huxleyi* proteome (*q*-value < 0.05, Supplementary Data 6), then grouped those domains by biological role (Fig. 2). This indicated that there have been innovations in the cytoskeleton and vesicular transport, ion transport, carbohydrate active enzymes, and proteins involved in calcium binding. Clearly, proteins of these categories have the potential to be involved in coccolithogenesis.

## Proteomic analysis of Ca-starved cells induced to form coccoliths by Ca replenishment

To investigate the protein machinery of coccolithogenesis we collected six independent proteomic datasets. Our entry point to this was a recalcification time course experiment where we made use of the phenomenon that cells grown at low calcium (low-Ca) concentrations (0.1 mM) do not produce coccoliths, but when returned to standard calcium (std-Ca) concentrations (10 mM) they rapidly start to calcify, and the first coccoliths appearing on the cell surface in 1 h, as we showed previously[20]. We used quantitative whole-cell proteomics to monitor changes in protein abundance over a 6-h time course of recalcification (Fig. 3a). The short timescale aimed to capture changes related primarily to coccolith formation, rather than secondary effects due to calcium as a nutrient. We compared recalcifying cultures to cultures maintained in a low-Ca medium over the same time period to take circadian and time-of-day effects into account. Despite many proteins changing in abundance over the 6-h period in both the non-calcifying (116 proteins) and recalcifying (308 proteins) cells, no proteins differed significantly in abundance between the two treatments at any time point (LIMMA adjusted *p*-value < 0.05, Supplementary Data 1).

To better understand this counterintuitive result, we asked whether the molecular program for coccolith formation continues at low calcium and whether the organic molecules associated with coccoliths continue to be secreted even in the absence of detectable calcification. The coccolith-associated polysaccharide (CAP) is a good marker for the organic constituents of the calcification program since histochemical work has shown it to be located inside the coccolith vesicle during coccolith formation, where it becomes tightly associated with the calcite and is eventually secreted with the mature coccolith[16]. We therefore analyzed the polysaccharides secreted by low-Ca and std-Ca grown cells. We added EDTA to the cultures to solubilize calcite and calcium-bound extracellular polysaccharides and immediately removed the cells by centrifugation.

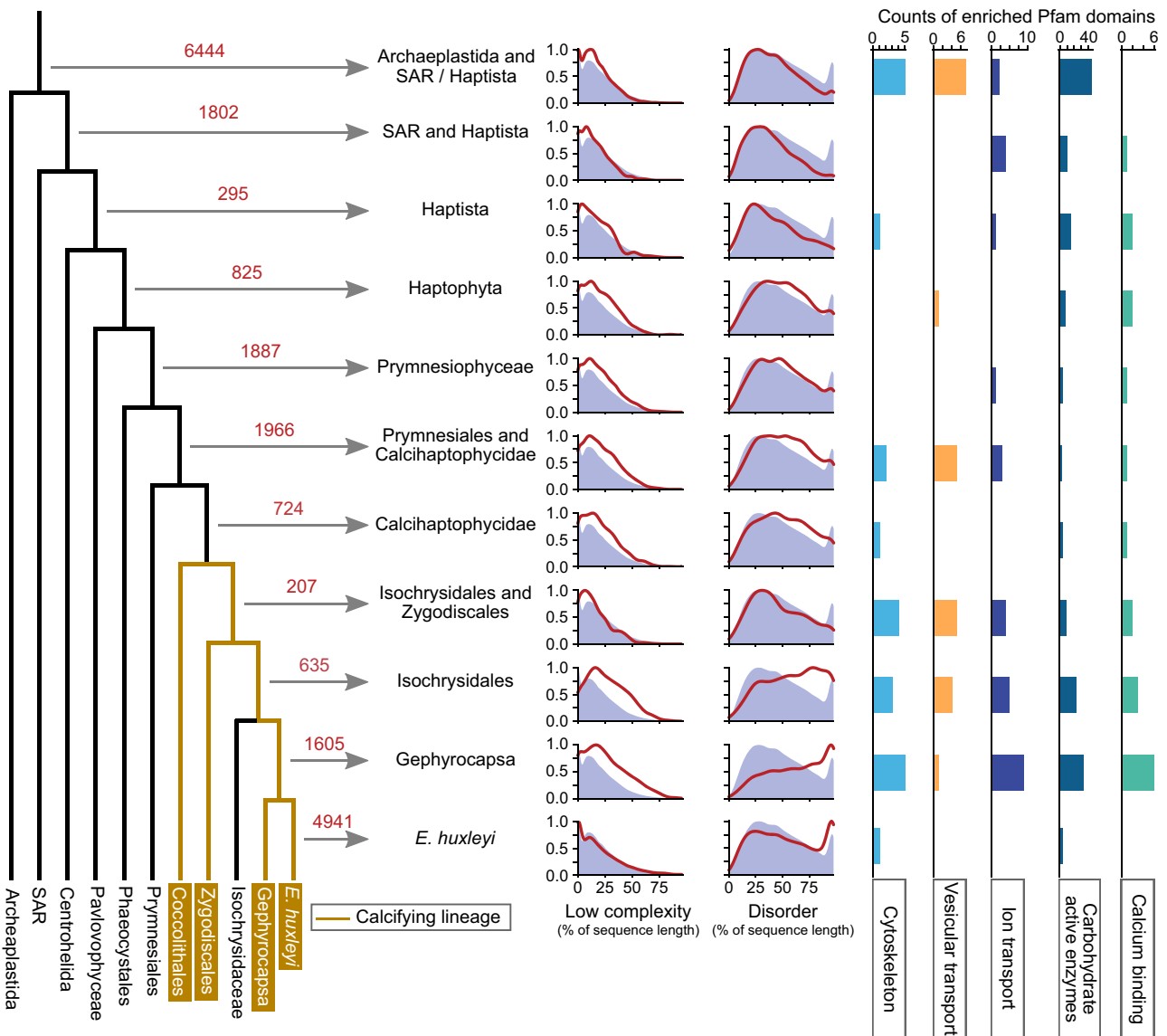

**Fig. 2 | Phylostratigraphic analysis of new *E. huxleyi* proteome.** Orthogroups were calculated for predicted proteomes of 38 species, including *E. huxleyi*, covering the phylogenetic range shown in the schematic tree. Red numbers indicate the number of orthogroups containing *E. huxleyi* sequences, that arise at each branch of the tree. For example, 825 orthogroups with *E. huxleyi* sequences arose in the common ancestor of all haptophytes. For each group, density plots of the proportion of each protein sequence that is low complexity and the proportion that is disordered are shown as red lines, while the purple background indicates the distribution for the entire *E. huxleyi* proteome. Pfam domains enriched in each orthogroups relative to the entire proteome were collated into categories and the counts of enriched domains for each category is displayed.

The supernatant was desalted by ultrafiltration and analyzed for monosaccharide composition by HPAEC-PAD and by SDS-PAGE. The HPAEC-PAD analysis showed that the monosaccharide composition and their relative quantities of the EDTA-soluble extracellular material (ESOM) were remarkably similar between the two calcium conditions and to the CAP solubilized from isolated coccoliths (Fig. 3b, Supplementary Fig. 6). The higher ratios for uronic acids and rhamnose for the CAP sample suggest that ESOM contains additional polysaccharides that are poor in these monosaccharides. The SDS-PAGE analysis showed that the running behavior of the dominant polysaccharide species and Stains-All staining pattern of the ESOMs were virtually identical to that of CAP (Fig. 3c). The combination of the proteomic and polysaccharide data, along with the speed of the recalcification response, support the hypothesis that the calcification machinery continues to be produced under low calcium conditions, despite an inability to produce calcite due to insufficient $Ca^{2+}$ availability.

## Proteomics of isolated coccoliths reveals compositionally biased proteins

We hypothesized that proteins may become associated with the coccolith during its synthesis, such that even after exocytosis, they remain occluded in the calcite, between adjacent crystals or in the CAP coat on the calcite surface. To identify such proteins (dataset 2), we isolated *E. huxleyi* coccoliths and performed washing with hot SDS and TritonX-100 followed by protease treatment to remove proteins that became associated with the coccolith surface during coccolith isolation (Supplementary Fig. 7). The coccoliths were then resuspended in EDTA solution to dissolve the calcite, and the solubilized organic material chemically deglycosylated to remove the CAPs, as these interfere with proteomics analysis. SDS-PAGE and silver staining of the deglycosylated material revealed a prominent band at ~40 kDa, which was protease sensitive and therefore represents a protein, as well as a smear across most molecular weights that likely represents degradation products (Fig. 4a). The deglycosylated material was digested in

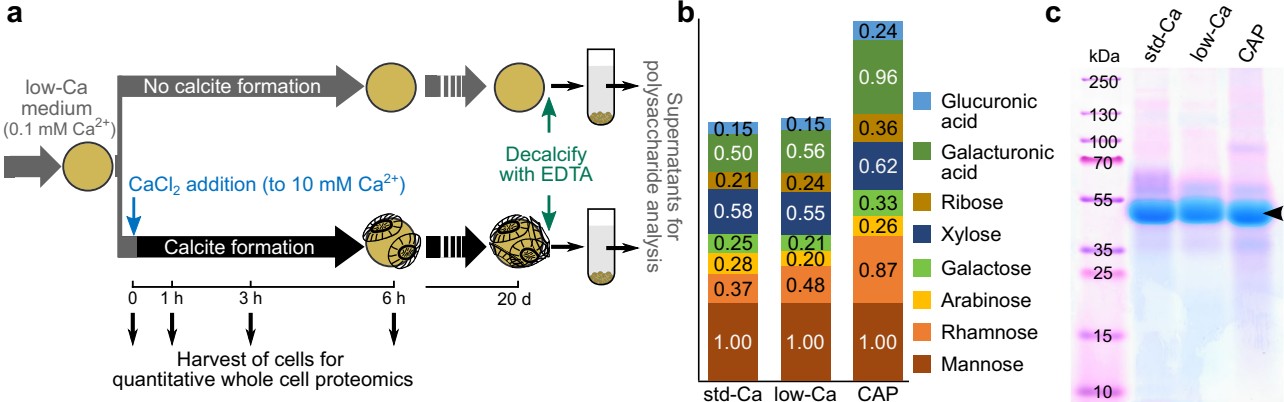

**Fig. 3 | CAP biosynthesis is maintained during low calcium growth.**
**a** Experimental design for probing CAP biosynthesis in low calcium grown cells. Cells grown in low calcium medium (low-Ca), in which they do not produce calcite, were aliquoted into two cultures. To one aliquot, calcium was added to induce calcite formation (std-Ca), while the other aliquot was continued in a low-calcium medium. At 0, 1, 3, and 6 h after calcium addition, samples were taken for microscopic and proteomics analyses. Extracellular polysaccharides were isolated from stationary phase cultures, as described in the materials and methods. The low-Ca culture for polysaccharide isolation was started with cells that had been propagated in a low-Ca medium for 1.5 months. The recalcification experiment was performed with cells that had been in a low-Ca medium for 14 days.

**b** Monosaccharide composition of the acid-hydrolyzed extracellular polysaccharide samples of low-Ca and std-Ca grown cultures and CAP extract from isolated coccoliths expressed as molar ratio as determined by HPAEC-PAD analysis. The numbers give the molar ratio of each monosaccharide to mannose, which was the most abundant monosaccharide in the main CAP of *E. huxleyi*. Note that not all monosaccharides that were detected could be quantified. For chromatograms see Supplementary Fig. 6. **c** SDS-PAGE analysis (*n* = 3, *n* = biologically independent samples) of extracellular polysaccharide samples and CAP extract stained with the cationic dye Stains-all. Arrowhead marks the dominant CAP of *E. huxleyi* coccoliths. Source data are provided as Source Data file.

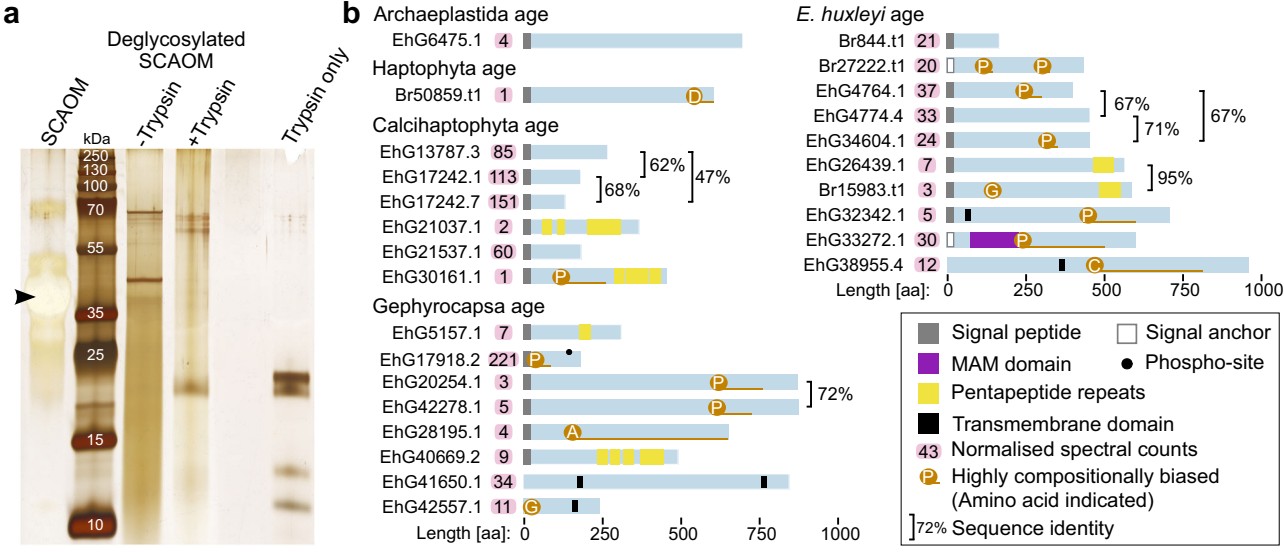

**Fig. 4 | Identification of coccolith proteins. a** Silver-stained Tricine-SDS-PAGE (*n* = 3, *n* = biologically independent samples) of soluble coccolith-associated organic material (SCAOM) isolated from purified coccoliths (Supplementary Fig. 7) and chemically deglycosylated SCAOM without further treatment (−Trypsin) and

after treatment with trypsin (+Trypsin). The arrowhead marks the dominant CAP of *E. huxleyi*. Note that unlike Stains-all staining (Fig. 3c), the CAP appears as a negative band in a silver-stained gel. **b** Schematic primary structures of the COPROs.

solution and subjected to mass spectrometry, identifying 82 protein groups at a posterior error probability ≤0.01, and in at least two of the three independent biological replicates. Fourteen proteins were excluded as likely contaminants based on homology to proteins of known function in evolutionary conserved processes.

The mechanisms of protein targeting coccolith vesicles are unknown, but given that there is ultrastructural evidence for the coccolithophore genus *Pleurochrysis* that CVs originate from the Golgi[14], it seems likely that luminal CV proteins should contain a signal peptide for import into the endoplasmic reticulum, from where they are transported into the Golgi system and from there into the CV. In total, 23 of the identified proteins had predicted signal peptides (21) or signal

anchors (2). Three proteins without signal peptides and one with a signal peptide have a predicted transmembrane domain that may target them into the CV membrane. These 26 proteins were considered candidate COccolith PROteins (COPROs) (Fig. 4b, Supplementary Data 2). We currently lack the computational tools to predict non-canonical secretory targeting pathways, and our knowledge of general cell biology in the haptophytes is furthermore extremely limited, meaning that it is entirely possible that some of the remaining 42 proteins are targeted to the CV. With this in mind, we consider these proteins as an expanded pool of candidates (Supplementary Fig. 8) and conservatively restrict COPROs to candidates with signal sequences.

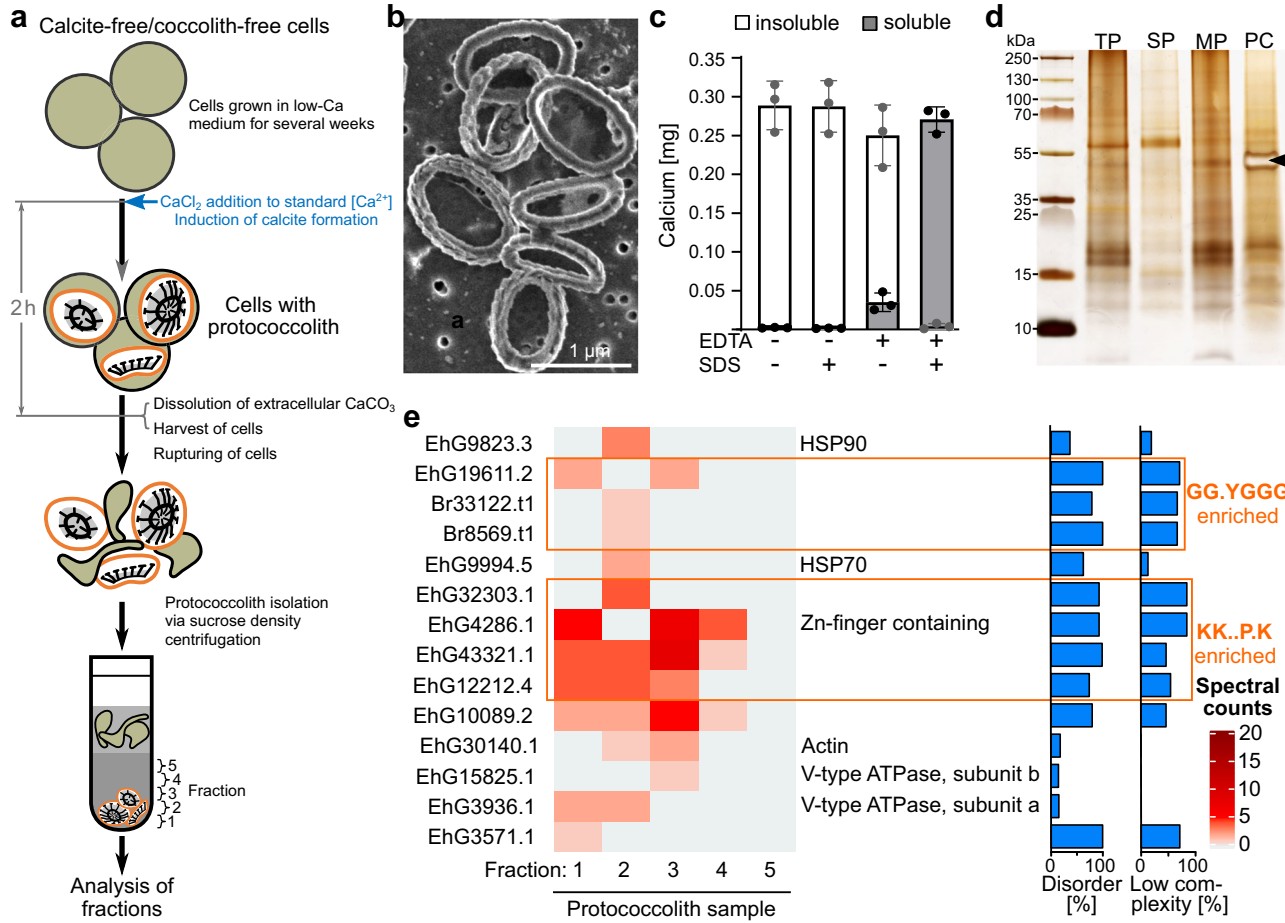

**Fig. 5 | Enrichment and proteomics analysis of proto-coccoliths. a** Experimental setup for the enrichment of proto-coccoliths. Cells were ruptured in a French press and aggregates and unbroken cells were removed by low-speed centrifugation. Proto-coccoliths were enriched by two consecutive rounds of sucrose density centrifugation. The last five fractions from the bottom were proteomically analyzed. **b** SEM micrograph of the material in the bottom fraction, showing that it contains proto-coccoliths. ($n = 2$, $n =$ independent biological replicates). **c** Fractionation of calcium in suspensions of proto-coccoliths treated with (+) and without (−) SDS and EDTA, determined by ICP-OES. Isolated proto-coccolith calcite is protected from dissolution by the calcium chelator EDTA. After the addition of SDS detergent, which solubilizes membranes, proto-coccolith calcite is dissolved by EDTA. Data are represented as mean ± SD ($n = 3$, $n =$ independent biological replicates). **d** Silver-stained SDS-PAGE gel of protein extracts ($n = 3$, $n =$ independent biological replicates) from whole-cells (TP), soluble proteins (SP), membrane-bound proteins (MP), and enriched proto-coccoliths (PC). The arrowhead marks the main CAP of *E. huxleyi*, which associates with the proto-coccolith calcite in the CV. **e** Distribution of proteins across the bottom fractions of a proto-coccolith. For each protein, the percentage of the sequence that is disordered and the percentage that is of low complexity is given. Proteins enriched in specific motifs are boxed.

In total, seven of the COPROs have conserved domains. In those of haptophyte age or younger, six contained pentapeptide motifs, suggesting that this structural element with unclear general function[43] may also have a role in the cell biology of calcification. One COPRO contained a MAM domain, which is typically an extracellular adhesive domain[44].

Some of the most abundant COPROs (based on normalized spectral counts) were Calcihaptophycidae age and Gephyrocapsa-age proteins of unknown function (Fig. 4b). These included a phosphorylated proline-rich Gephyrocapsa-age protein (EhG17918.2) and two Calcihaptophycidae-age proteins (EhG17242.1, EhG17242.7). There are various candidates for the identity of the strong band at ~40 kDa in the SDS-PAGE. In particular, the proline-rich protein Br27222.t1 and a trio of homologous *Emiliania*-age proteins (EhG4764.1, EhG4774.4 and EhG34604.1) may contribute to the band based on their predicted molecular weights. MS spectra of digests from the gel-separated material were poor, so the identity of the 40 kDa band could not be reliably confirmed. EhG4764.1 was the only one of the trio which was detected in the gel-separated material (Supplementary Fig. 9). Another candidate for the band at 40 kDa is the Gephyrocapsa-age EhG17918.2, which was detected in gel-digests and in-solution digests.

It was striking that six COPROs (EhG13787.3, EhG17242.1, EhG17242.7, EhG21037.1, EhG21537.1 and EhG30161.1) were in the set of 254 proteins of Calcihaptophycidae age lost in non-calcifying *Isochrysis*.

## Proteomics of proto-coccoliths isolated from the cytosol

We attempted to isolate CVs in order to more directly identify proteins involved in coccolith formation (dataset 3). Cells were grown in a low-calcium medium for several weeks such that all cells lacked coccoliths, and then calcite formation was induced by adding calcium back to the medium. Before any coccoliths appeared on the surface, cells were lysed and subject to density gradient centrifugation, resulting in calcium-rich fractions at the bottom of the gradient (Fig. 5a, Supplementary Fig. 10). Scanning electron microscopy (SEM) analysis of the bottom fraction revealed immature coccoliths (Fig. 5b), and cryo-SEM analysis of freeze-fractured bottom fractions showed organic material enclosing the calcite (Supplementary Fig. 11). The calcite was protected from dissolution with EDTA unless the proto-coccoliths had prior treatment with detergent (Fig. 5c). Altogether, these data suggest that the enriched proto-coccoliths are bound by a membrane, which for simplicity we refer to as CV membrane here. In *Emiliania*, however, the CV membrane includes not only the membrane in contact with the

calcite crystals but also that of the reticular body, an organelle closely associated and partly fused with the CV[45]. For simplicity, the corresponding proteomics data will be referred to as the CV dataset. In SDS-PAGE, SDS solubilized organic material of the enriched proto-coccoliths displayed a band pattern distinct from that of whole-cell extracts, with a prominent band at ~50 kDa, apparently representing the CAP (Fig. 5d), which previous immunohistochemistry work showed to be associated with proto-coccoliths[16]. The fractionation procedure was carried out with calcifying cells and cells of a non-calcifying mutant of the calcifying strain AWI1516 (mutant strain CCMP2090, N-cells) as a negative control for comparison, and the proteins in each fraction were identified by mass spectrometry. Having removed clear contaminants, 14 proteins enriched in the bottom fractions of the proto-coccolith samples remained (Fig. 5e, Supplementary Data 3). Among them were V-type ATPase subunits (EhG3936.1, EhG15825.1), consistent with previous reports in the literature from a study attempting to isolate CVs from *Pleurochrysis carterae*[32]. It has been hypothesized that the V-type ATPase could provide a proton gradient to drive $Ca^{2+}$ loading of the CV prior to calcification[35]. However, since no data are yet available on the in vivo localization data of this enzyme in *Emiliania*, the presence of ER membrane, which is closely associated with the CV[45] could also explain its presence in our proto-coccolith preparations. We also identified actin (EhG30140.1), which may be involved in shaping the CV and coccolith exocytosis, and one (EhG9823.3) out of ten HSP90-like proteins in the proteome (Supplementary Table 4). EhG9823.3 lacks a signal peptide, so it is probably not a luminal CV protein. Instead, its association with extracellular coccoliths could be due to binding to the cytosolic side of the CV membrane. Mammalian HSP90 has been shown to interact with phospholipid membranes[46], mediating deformation and release of exosomes[47], which could be consistent with a role for EhG9823.3 in coccolith vesicle membrane dynamics. In addition, we identified eight proteins of unknown function, with high predicted intrinsic disorder, and low sequence disorder (Fig. 5e).

## The coccosphere is rich in diverse proteins, many of unknown function

It is conceivable that some CV lumen proteins with a role in coccolith maturation might not end up within the calcite or tightly bound to the coccolith, but might be released into the coccolith-enclosed microenvironment surrounding each cell, called coccosphere, during coccolith exocytosis. For this reason, we decided to examine the proteome of the coccosphere (dataset 4). It is important to note that we also expect proteins unrelated to calcification, such as proteins involved in nutrient uptake or biotic interactions to be present in the coccosphere, so orthogonal lines of evidence are crucial to the interpretation of this experiment. To get rid of proteins secreted into the medium, we pelleted calcified cells and resuspended them in fresh medium. The coccosphere was then dissolved by an EDTA treatment to solubilize associated proteins (Supplementary Fig. 12). The cells, still intact, were then removed by centrifugation, and the supernatant was analyzed by protein mass spectrometry. An identical procedure was applied to cells of the same strain grown under low-Ca conditions, where they do not produce coccoliths and thus lack a comprehensive coccosphere. We required proteins to be identified in at least three of the four replicates, at a posterior error probability ≤ 0.01 for inclusion. This identified 110 proteins, 15 of which were judged to be contaminants. Of the remaining proteins, 78 were exclusively from the calcifying condition (std-Ca), while 17 were also identified in the low-Ca samples (Supplementary Data 4). There were no proteins only identified in low-Ca.

The std-Ca specific proteins were considered putative coccosphere proteins and most of the spectra recorded from these were from proteins of haptophyte age or younger (Supplementary Fig. 13). Of the 78 proteins, 41 had conserved domains (Supplementary Fig. 14)

and 12 had transmembrane domains (Supplementary Figs. 14 and 15), and a subgroup (Fig. 6a) contain domains immediately suggesting a role in coccolith biogenesis and coccosphere assembly. These include cytoskeleton proteins (myosin, actin, kinesin), a carbonic anhydrase, glycosyl hydrolases, ion channels (a bicarbonate transporter, an SLC24 type $Na^+/Ca^{2+}K^+$ exchanger, $Mg^{2+}$ transporter, and a chloride channel), and an ABC transporter. The reproducible detection of transmembrane proteins in our coccosphere material suggests the presence of extracellular vesicles within the coccosphere, possibly originating from the plasma membrane. Indeed, recent work has found that *Emiliania* releases extracellular vesicles[4]. Other domains that occurred in multiple proteins include ShK-like domains (five proteins) and PH domains (two proteins), but the roles of these in the context of coccolith formation are unclear. The MAM domain containing protein EhG33272.1 was also found among the COPROs, and the actin EhG30140.1 in the CV dataset (Fig. 4b, Supplementary Fig. 14). The 37 proteins without conserved domains contain some sequence biases and in particular P-rich regions (Fig. 6b, Supplementary Fig. 15). The *Gephyrocapsa*-age P-rich protein EhG42278.1 was also found among the COPROs (Fig. 4b).

## Overlap between datasets identifies candidates for proteins involved in coccolith formation

We collected two additional datasets (datasets 5 and 6) to provide orthogonal support for proteins identified so far. This included a quantitative comparison of whole-cell protein levels in the day and at night, since *E. huxleyi* calcifies much more rapidly in the light than in the dark[48,49], in which we found 142 proteins to have changed in abundance (LIMMA analysis, adjusted $p$-value < 0.05). Second, we examined proteins that varied in abundance between C-cells and non-calcifying cells of CCMP2090 by whole-cell proteomics, in which we found 1291 proteins to be changing (LIMMA analysis, adjusted $p$-value < 0.05) (Supplementary Data 5). Note that we do not attach particular significance to the direction of regulation in these experiments, since factors important in calcification could be either positive or negative regulators of it and since the nature of the calcification defect in N-cells is unknown. The overlap between all the datasets collected was then examined. In total, 114 proteins displayed overlap, but 86 of these were only in the C-cell vs N-cell data and the Light vs Dark data (Fig. 7, Supplementary Data 5). However, 19 proteins displayed overlap between the C-cell vs N-cell data and the coccosphere data (including three proteins additionally present in the Light vs Dark dataset), two between C-cell vs N-cell data and the COPRO (including one protein additionally present in the Light vs Dark data), and two between the COPROs and the coccosphere (Fig. 7, Supplementary Table 5).

Using the presence of a subcellular targeting motif and the type of experiment identifying the proteins as key parameters, we developed a model for the involvement of the proteins that overlap between datasets in coccolith formation (Fig. 8).

## Discussion

The coccolithophore *Emiliania huxleyi* has been the principal model organism for studies of haptophyte biology and for investigating the mechanisms of $CaCO_3$ biomineralization, a characteristic aspect of coccolithophore biology. This biomineralization process has consequences for the global carbon cycle because it impacts the exchange of $CO_2$ between the ocean and the atmosphere and the burial of carbon on the ocean floor. It is therefore critical to know more about the biology of this process to be able to understand how coccolithophores will respond to future changes in ocean chemistry and climate in order to make more accurate predictions about the future of the carbon cycle and the impacts of climate change. The genome of *Emiliania* was the first haptophyte genome to be published, largely annotated with de novo tools trained on organisms only very distantly related to

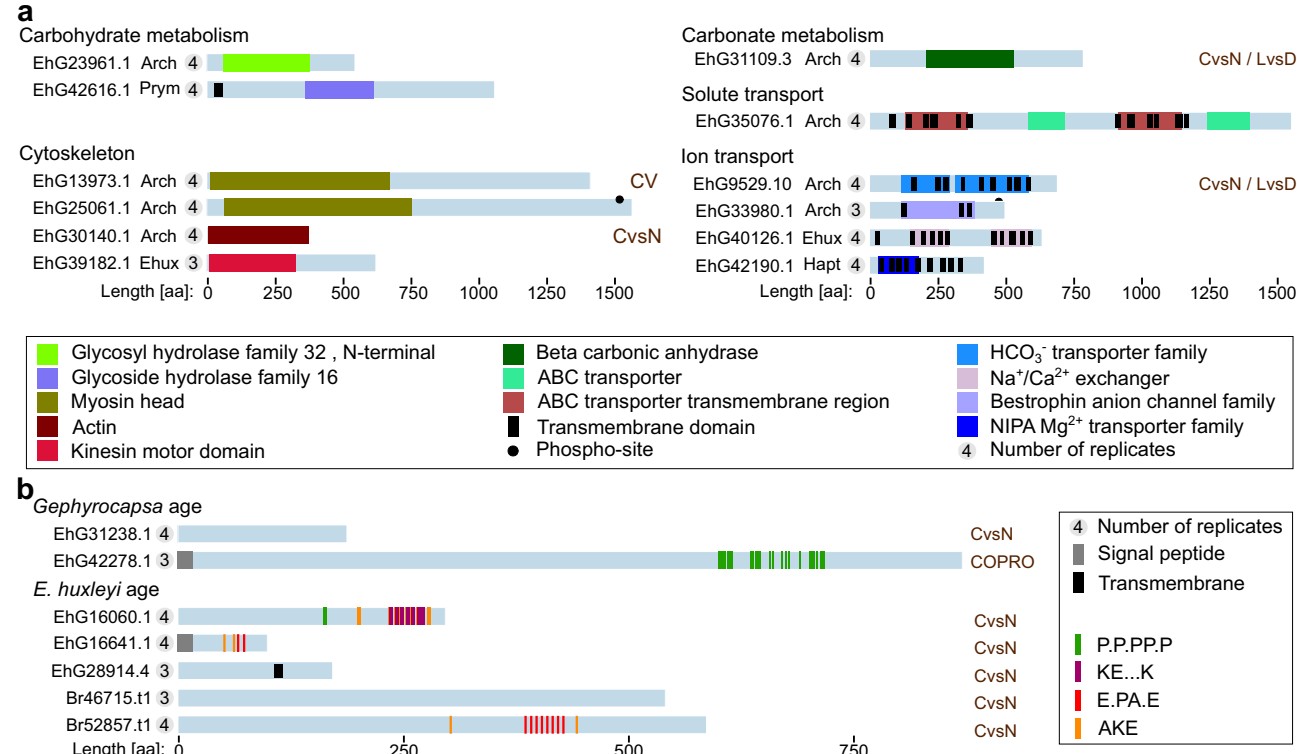

**Fig. 6 | The proteome of the coccosphere. a** Schematics of a selection of putative coccosphere proteins that may have a role in calcification, identified in at least three replicates and with conserved domains. **b** Schematics of putative coccosphere proteins without conserved domains, also identified among our other datasets (see below for CvsN dataset). Overlaps with other datasets are indicated (CvsN C-cells vs N-cells dataset, LvsD Light vs Dark dataset, CV coccolith vesicle, COPRO coccolith protein) as are sequence motifs enriched in the coccosphere proteins relative to the entire proteome.

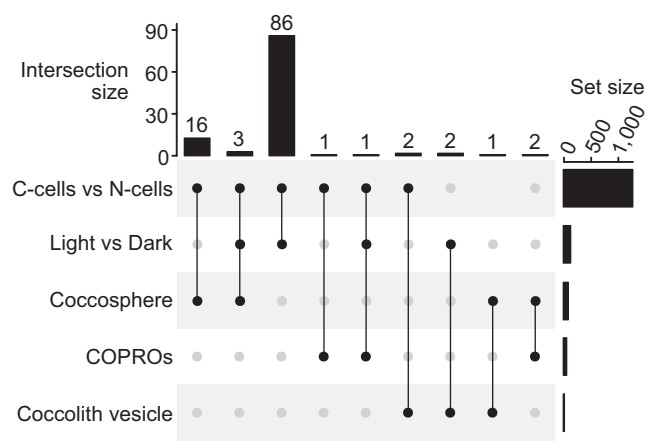

**Fig. 7 | Overlap between datasets.** 'C-cells vs N-cells' and 'Light vs Dark' datasets are proteins regulated in either direction, $q$-value < 0.05. Coccosphere proteins are Std-Ca-specific proteins identified in at least two replicates by at least two peptides. COPROs are as described above. CV proteins are those found exclusively in the C-cell density gradients.

haptophytes[27]. We hypothesized that this might be limiting for our understanding of unique aspects of coccolithophore biology, such as coccolith formation. In particular, some of the genes and proteins that mediate the process of coccolith production are likely to be unique to this group of organisms and may be poorly predicted by ab initio gene prediction methods. Here we overcome these limitations, using long-read transcriptome sequencing in combination with a new *E. huxleyi* genome to generate an improved predicted proteome. This has allowed us to perform a suite of proteomic experiments investigating

the molecular machinery of calcification, identifying many more proteins than would previously have been possible. For example, of the 26 COPROs we identify, only eight have identical sequences in the Emihu1 proteome. Other genes 'missing' in Emihu1 can now be found in Emihu2. For example, the nitrate reductase gene was missing, although *E. huxleyi* has experimentally verified nitrate reductase activity[50], it can be found in Emihu2. Furthermore, of 19 flagellar proteins found to be missing by von Dassow et al.[8], 10 are present in Emihu2 (Supplementary Table 6), and seven classes of myosins that had not been previously been recognized in hapto-phytes (Supplementary Fig. 16).

Several of the proteins we identify have conserved domains suggestive of particular roles in calcification. For example, we found several ion transporters in our data (Fig. 6, Supplementary Note 7). The best candidate for bicarbonate uptake is EhG9529.1: a bicarbonate transporter, found in the coccosphere, C-cells vs. N-cells and light vs dark data. EhG31109.3 was identified in the same set of experiments and encodes a beta-carbonic anhydrase, which could be involved in regulating bicarbonate availability for calcification. The diploid Emihu2 genome contains 15 loci putatively encoding $Ca^{2+}/H^+$ exchangers (CAX proteins, Supplementary Table 14), of which one is represented in our proteomics data. Previous transcriptome analysis has found an SLC24-type $Na^+/Ca^{2+}K^+$ exchanger to be specific to the diploid, calcifying life stage of *E. huxleyi*[34] and we also found the SLC24 transporter EhG40126.15 in the coccosphere data (Supplementary Fig. 17, Supplementary Data 4). We found 10 subunits of the V-type ATPase in our proteomic data, nine of which were upregulated in N-cells relative to C-cells, and two of which were in the CV data. This supports previous findings that V-type ATPases may be in the CV membrane[32] and may suggest that "over-acidification" of CV could be responsible for the lack of calcification in the N-cells. In support of this idea, an N-cell strain has been reported not only to produce the CAP

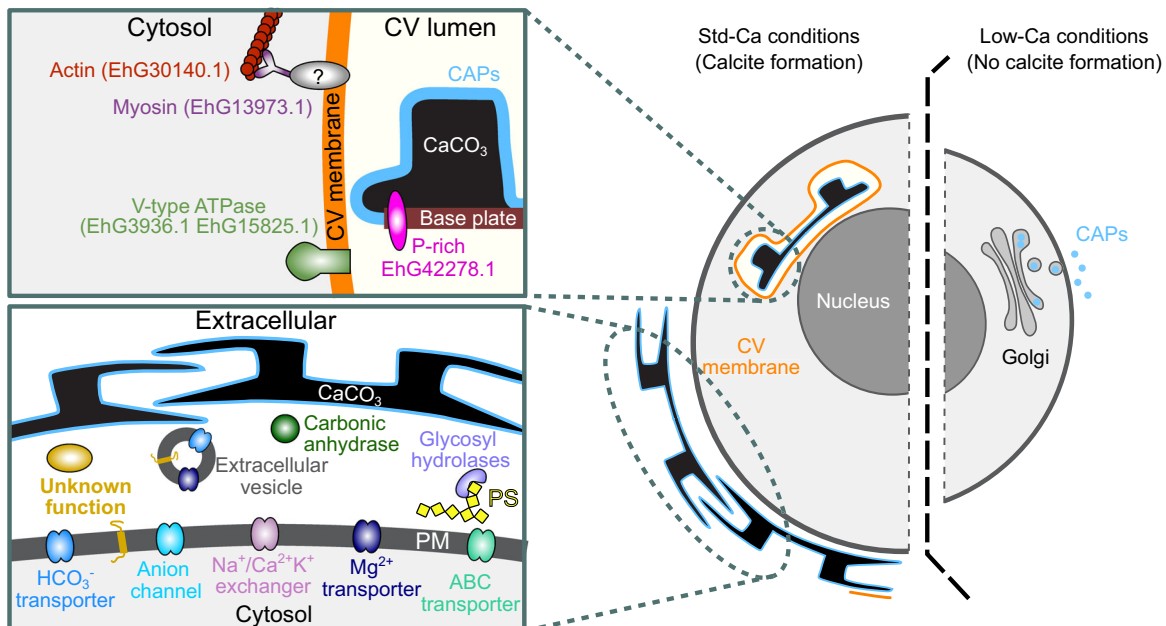

**Fig. 8 | Model for the involvement of proteins in coccolith formation.** Proposed molecular organization of the CV and the coccosphere. At standard calcium concentration (Std-Ca), coccoliths form inside the specialized CV, where calcite nucleation and morphogenesis are controlled by luminal polysaccharides (CAPs, light blue) and proteins, as well as cytosolic proteins interacting with the CV membrane (orange). The inset (top left) shows the hypothesized distribution of selected proteins identified in two or more proteomic experiments inside and outside the CV and their proposed function. An intriguing candidate CV luminal protein is EhG42278.1 which possesses multiple Ser-(Pro)$_{3-4}$ peptide repeats, defining structural features of glycosylated algal and plant cell wall proteins known as extensins[94], which play a crucial role in cell wall assembly[95]. EhG42278.1 may play a comparable role to the extensins in the assembly of the base plates. Other strong candidates for CV luminal proteins are the double pentapeptide motif-containing proteins EhG21037.1 and EhG40669.2, EhG33272.1, and EhG6475.1. V-type ATPase

in the CV membrane may regulate the pH in the CV lumen, and the cytosolic proteins actin and myosin may be involved in CV morphogenesis and coccolith extrusion. The coccosphere (inset bottom left) is an environment rich in proteins, some of which, such as carbonic anhydrase, may have roles in coccolith biogenesis. Most of the coccosphere candidate proteins (17 in total, 16 globular and one transmembrane protein) are of unknown function. The exchange of ions between the cell and the environment is facilitated by a plethora of ion transporters and channels, of which five were identified here (HCO$_3^-$ transporter: EhG9529.1, Anion channel: EhG33980.1, Na$^+$/Ca$^{2+}$ K$^+$ exchanger: EhG40126.1, Mg$^{2+}$ transporter: EhG42190.1, ABC transporter: EhG35076.1). At low calcium concentrations (right side), calcite formation ceases, but not CAP biosynthesis and secretion, and therefore we propose that extracellular calcium availability is not an activator of the genetic program underlying coccolith formation.

but also to contain a CV and an associated membranous structure called the reticular body[51,52].

The cytoskeleton is very likely to be critical for orchestrating the process of CV development and calcite morphogenesis[53]. Indeed, we found actin (EhG30140.1) in both the coccosphere and CV data and studies have previously shown an increase in malformed coccoliths on treatment with microtubule and actin depolymerizing drugs[54]. It is interesting that the myosin VI class proteins EhG31424.1 and EhG25061.1 were found in the coccosphere data. Myosin VI is the only myosin that moves toward the minus end of actin[55] and is generally associated with endocytic vesicles and endosomes[56,57], so both proteins may play a role during coccolith exocytosis and subsequent recovery of the CV membrane.

The fact that many of the proteins identified were of unknown function was to be expected given the lack of well-studied close relatives of *E. huxleyi* and that coccolith formation is unique to this group of haptophytes. These novel proteins deserve further study, to determine if they play roles in calcification. The only specific *E. huxleyi* protein previously proposed to have a role in the nucleation and control of crystal growth is GPA (EhG32816.1)[31]. This acidic protein was identified in trichloroacetic acid extracts of cells containing CAP precursor molecules[31]. Subsequent immunofluorescent assays demonstrated that GPA is not associated with coccoliths but is present in a layer on the cell surface that is not removed by decalcification[31]. This localization is consistent with the absence of the protein from our coccolith or coccosphere-associated protein samples. We did detect GPA (2

peptides, 5 spectra) in the recalcification time course experiment, but not in any other dataset, providing the first protein-level evidence of its expression since the original publication. The expression of the *GPA* gene has been shown to be correlated negatively with calcification[58], and in our opinion, the previously identified correlation between coccolith morphotype and genetic variation at the *GPA* locus[59] is likely not causal.

A key advantage of our data is that the orthogonal nature of the experiments provides independent evidence for the role of proteins in calcification. Yet more evidence can be gathered by comparing our protein-level data to transcript-level data from the literature. Von Dassow and coworkers compared gene expression in calcifying, diploid *E. huxleyi* with non-calcifying, haploid *E. huxleyi* in two separate studies via EST sequencing[8,34]. By searching the EST consensus sequences against our new transcriptome, we identified large overlaps between this data and ours (Supplementary Fig. 18). For example, 245 of the proteins we identified to be differentially expressed between C-cells and N-cells were also differentially expressed at the transcript level between haploid and diploid *E. huxleyi*, and 12 of the COPROs also overlapped with this data. This provides further reassurance that a high proportion of the proteins we identified are possibly related to calcification.

An intriguing outcome of this study was the molecular complexity of the coccosphere. As well as being rich in CAP, it seems to house a wide variety of proteins, including enzymes potentially involved in redox processes, detoxification, protein folding, carbohydrate metabolism and inorganic carbon acquisition. This suggests that the

coccosphere may have impacts on cellular physiology and fitness beyond those endowed by the coccoliths themselves.

The new *E. huxleyi* genome, transcriptome and predicted proteome described here along with six proteomic databases relating to calcification, provide several excellent candidates for the protein machinery underlying coccolithogenesis. These data are also useful for studying various other aspects of coccolithophore biology, as they currently provide the most comprehensive window into the functional potential of this group of algae at the molecular level.

## Methods

### Algal culture

*Emiliania huxleyi* strain AWI1516, which produces coccoliths, and its closely related sister strain CCMP2090, which does not produce coccoliths, were grown in artificial seawater medium Aquil, at 18 °C and under 50 μE light in a 12/12-h light/dark cycle. The concentration of nitrate in the medium was 0.2 mM and of phosphate 10 μM. Under standard growth conditions, where the cells form calcite, the medium contained 10 mM $CaCl_2$ (std-Ca medium), while under low-Ca growth conditions, where the cells do not form calcite, it contained 0.1 mM $CaCl_2$ (low-Ca medium). In experiments with cells from low-Ca growth conditions, the cultures had been kept in a low-Ca medium for at least 2 weeks before use. For cells inoculated into low-Ca medium, the extracellular coccoliths were dissolved with EDTA, and cells were washed with Ca-free medium before inoculation, as previously described[20]. Algal cell density was measured using a Beckman Coulter Z2 particle counter.

### DNA extraction, library preparation and SMRT sequencing

For the extraction of high molecular weight DNA, cells were disrupted using the Tissuelyser II (Qiagen) and Lysing Matrix E (MP Biomedicals) and DNA was purified using the NucleoSpin Food kit (Macherey-Nagel). DNA quality was assessed using the Agilent FEMTOpulse system and DNA quantity was measured using the Thermo Scientific Qubit system. PacBio libraries were prepared from unfragmented DNA with the SMRTbell Template Prep Kit 1.0 (Pacific Biosciences) followed by size selection for enrichment of fragments ≥10 kb (Blue Pippin, SAGE Science). After purification with magnetic beads, the library was sequenced on the Sequel with Sequel DNA polymerase and binding kit version 3.0 and sequencing chemistry version 3.0 for 1200 min. Data was collected from two SMRT cells, with a total output of 8.86 Gb (Supplementary Note 1).

### Genome assembly

The continuous long-read (CLR) sequencing data were processed using SMRTLink v7.0.1, which relies on the HGAP assembler. Falcon unzip[60] (v1.3.5) was used to generate an initial phased assembly but only resolved a limited degree of the heterozygosity in the genome (only 10% of the sequence was in haplotigs), so we applied the Purge Haplotigs pipeline[61] (v1.1.0) and optimized parameters to generate an improved haploid assembly (see Supplementary Fig. 21).

### RNA extraction, library preparation and PacBio Isoseq sequencing

For RNA extraction, cells were harvested in the logarithmic growth phase by vacuum filtration on autoclaved Whatman glass microfiber filters (grade GF/A). The filters with the cells were washed with 1 mL fresh medium, transferred to 2 mL Eppendorf tubes, frozen in liquid $N_2$ and stored at −80 °C until further processing. Harvesting a culture this way took less than 1 min. Total RNA from each sample was isolated using a Direct-zol RNA purification kit (Zymo Research) and RNA integrity and quantity were assessed using the RNA Nano 6000 Assay Kit and the 2100 Bioanalyzer System (Agilent). To obtain good coverage of the transcriptome and the repertoire of transcript isoforms, RNA was isolated for Isoseq sequencing from cells that had been (A)

6 h into the light phase and (B) 6 h into the dark phase in Ca-replete medium, (C) 6 h into the light phase in low-Ca medium, and (D) 6 h into the light phase after 2 days in zero-P-medium. Four independent biological replicates were pooled for sequencing post RNA extraction and quality control. Further sample preparation was according to the protocol "Procedure & Checklist - Iso-Seq Template Preparation for Sequel Systems", in which cDNA is synthesized with the SMARTer PCR cDNA synthesis kit (Clontech) and then reamplified by PCR using the KAPA HiFi PCR kit (Kapa Biosystems). SMRT bell libraries were constructed with the Pacific Biosciences SMRTbell Template Prep Kit 1.0 and sequenced with Sequel DNA polymerase, binding kit version 2.1 and sequencing chemistry version 2.1 for 10 h on the Sequel.

### Transcriptome assembly

Sequel data were processed with SMRTLink v6.0.0.47841 (v3.0.0 commit 1035f6f;–noPolish–minPasses 1). Data from each experimental condition were combined for downstream steps, which were performed with Isoseq v3.1.0. This included processing by LIMA (v1.6.1) for adapter trimming, Sierra v0.7.1 for the detection of full-length non-chimeric (FLNC) reads and clustering FLNC reads to isoforms, and finally tango v0.7.1 for polishing the isoforms. This pipeline was performed with default parameters, and isoforms called high quality (HQ) were used in subsequent steps. The HQ reads were first mapped against the published *E. huxleyi* organelle genomes (mitochondrial genome: NC_005332; plastidial genome: NC_007288). All unmapped isoforms were then mapped against the new *E. huxleyi* haploid and diploid genomes minimap2 (v2.17-r943-dirty; -ax splice:hq -uf -C1 -cs = long -N 4). Finally, we used TAMA Collapse[62] (-a 100 -z 250 -e longest_ends -x no_cap) to perform isoform clustering on the diploid genome and alignment, generating the final transcript and isoform sequences.

### Generating the predicted proteome

To ensure the predicted proteome was as complete as possible, we decided to use the diploid genome and associated gene models as the starting point. We felt this was particularly important given the level of heterozygosity in the genome. Given the presence of PacBio sequencing errors in both the genome and the Isoseq reads, both sequence sources were considered as candidates for ORF generation for each locus. ORFs were predicted using getorf[63], gmst[64] and TransDecoder[65], with equivalent parameters where possible (getorf -find 1–noreverse–minsize 120; gmst.pl–strand direct–faa–fnn–format GFF; TransDecoder.LongOrfs -m 40 -S -t). For the getorf output, the longest ORF per locus was extracted. Hmmscan[66] was run on all ORF outputs using the Pfam-A.hmm database (v32.0) and was used to refine the Transdecoder ORF selection (TransDecoder.Predict–retain_pfam_hits–single_best_only –no_refine_starts). For further analyses, only Pfam hmmer matches with an i-Evalue < $10^{-3}$ were used. ORF selection for each gene model then proceeded by the procedure detailed in Supplementary Note 2 (code and input data available at figshare (https://doi.org/10.6084/m9.figshare.20464293.v1)). We do not expect our transcriptome to cover the full expression potential of the organism, in particular, because we sequenced only diploid C-cells and not haploid S-cells, so ORFs from de novo gene prediction were also included. For gene prediction, repetitive regions of the diploid genome assembly were identified via RepeatModeler (Version: 1.0.11; Parameter: "-engine ncbi"; without RepeatScout) and masked using RepeatMasker (Version: 4.0.9; Parameter: "-xsmall -gff -nolow"). Gene prediction itself was carried out using the BRAKER[67] v2.1.0 pipeline (–gff–softmasking), with the masked diploid genome and the Isoseq HQ isoform mapping results as input. To select de novo genes not represented in the Isoseq data, we used gffcompare[68] (v0.10.6) to compare the TAMA gene models and the de novo predicted models from the Braker pipeline. Braker models were selected for ORF finding if they were substantially different from TAMA models, meaning that they fell into GFF compare

classes s, x, li, y, p, u, o, k, and j. The default ORF predicted by BRAKER was used, adding an additional 36,167 sequences to the final proteome.

## Database quality control

For details of input data and total protein sample preparation see Supplementary Note 5. Spectra in mgf format were de novo sequenced with PepNovo+ v 3.1[69,70] using the CID_IT_TRYP model and the option '-PTM M + 16:C + 57'. Spectra scoring less than a PepNovo score of 60 in their de novo solution were discarded, and the remaining spectra ('HQ spectra') were used to perform database searches against the relevant databases with comet using parameters as described in 'Proteomic Methods'. *E. huxleyi* spectra were searched both against the new Emihu2 database and against the JGI Emihu1 database ('Best proteins', https://phycocosm.jgi.doe.gov/Emihu1). *Arabidopsis* data were searched against the Araport11 database (https://www.arabidopsis.org/download)[71] (Supplementary Note 3). Databases were supplemented with a database of common MS and laboratory contaminants. After running Peptide Prophet (TPP version-5.2.0.1), the resulting pep.XML files were parsed to generate tables of the best peptide probability assigned to each spectrum. The HQ spectra were also clustered using the PRIDE Cluster algorithm[72] (spectra-cluster-cli-1.1.2.jar with "-filter immonium_ions").

After applying a peptide probability cutoff of 95%, the proportion of spectral clusters with database identification was calculated for each sample/database combination. The *Arabidopsis* data were provided to give the context of a good-quality database from a well-studied organism. To make the results comparable to the *E. huxleyi* data, the *Arabidopsis* results were sampled so that the spectra entering the calculation had a similar de novo sequencing score distribution to either the *E. huxleyi* total protein data or the coccolith-associated protein data, as indicated in Fig. 1g.

## Structural and functional annotation

The predicted proteome was annotated with InterProScan (InterPro 82.0)[73], EggNog-Mapper (v5.0)[74], HECTAR for localization[75] and pureseqTM (transmembrane helices[76]).

## Phylostratigraphic analysis

Apart from the *E. huxleyi* protein sequences (which derived from this work), all sequences used in the Phylostratigraphic analysis were taken from the EukProt database[77] which relies heavily on the Marine Microbial Eukaryote Transcriptome data[78]. The species included are detailed in Supplementary Table 3. Orthogroups were inferred using Orthofinder (v2.3.8)[79], with a user-specified guide tree (Supplementary Fig. 5). The full results of the Orthofinder analysis are available at figshare (https://doi.org/10.6084/m9.figshare.20463900). Pfam domains were identified using InterProScan (InterPro 82.0). Enriched domains were identified by calculating hypergeometric $p$-values from which $q$-values were estimated using the R function qvalue_trunc.R (https://rdrr.io/github/StoreyLab/qvalue/src/R/qvalue_trunc.R). A local fdr estimate <0.05 was required for a domain to be considered enriched.

## Analysis of over-represented motifs, sequence complexity and disorder

Over-represented motifs were discovered using motif-x[80] via the ProminTools suit[37]. Predicted disorder and low complexity were likewise calculated using the ProminTools suit, which relies on VLS2[81] and SEG[82] for these operations respectively. Regions of compositional bias were discovered using fLPS[83].

## Coccolith isolation

Coccoliths were isolated from calcifying 100 L cultures by boiling cells in 2% SDS/50 mM $NH_4HCO_3$ solution and after that with 1% TritonX-100/50 mM $NH_4HCO_3$ solution to remove membranes and soluble organic macromolecules. After each boiling step, the suspension was allowed to cool down to room temperature with vigorous stirring, and after pelleting the coccoliths by centrifugation, they were washed three rounds with 50 mM $NH_4HCO_3$ solution to remove the detergent. Afterward, the coccoliths were freeze-dried and stored at −20 °C until further usage. For proteomic analysis, isolated coccoliths were treated overnight at 37 °C with Pronase (0.1 mg/mL in 100 mM TRIS-HCl pH 7.6, 0.5% SDS buffer) to remove surface-bound proteins. Proto-coccoliths (CV isolation) were isolated as described in Supplementary Note 4.

## Preparation of EDTA-soluble organic material (ESOM)

ESOM from isolated coccoliths was prepared by resuspending dry coccoliths in 0.5 M EDTA pH 8 solution. The samples were incubated with constant mixing until the color of the suspension changed from whitish to translucent. Afterward, the remaining insoluble material was pelleted by centrifugation at $2500 \times g$ for 15 min. The supernatant, coccolith-associated ESOM, was transferred into a fresh tube and stored at −20 °C until further usage. Extracellular ESOM of cells was prepared by adding EDTA solution to a final concentration of 10 mM to early stationary phase cultures. Immediately after EDTA addition, the cells were pelleted by centrifugation (10 min, $2000 \times g$). This short-time EDTA exposure dissolved extracellular coccoliths, as was checked by light-microscopy, and solubilizes macromolecules from the cell surface, but leaves the cells intact. After centrifugation, the supernatant was passed through a 0.2-μm filter to remove particulate material that was not pelleted during centrifugation. The ESOM in the flow-through was concentrated and desalted by ultrafiltration (cut-off 5 kDa).

## Monosaccharide analysis

The isolated EDTA-soluble organic material from cells and coccoliths was hydrolyzed with 2 M trifluoroacetic acid (TFA) for 4 h at 100 °C followed by adding 2-propanol and evaporation at 40 °C. Afterward, the hydrolyzed material was washed three times with 2-propanol (evaporation at 40 °C), resuspended in distilled water and supernatants, containing the monosaccharides, were collected after centrifugation to pellet undisolve material. Monosaccharides composition and quantification were determined by HPAEC-PAD using a Carbopac PA20 column on an ICS6000 system (Thermo Scientific Dionex). Each sample was analyzed under two elution conditions. Elution condition I utilized an isocratic NaOH concentration of 2 mM to separate the neutral monosaccharides, followed by a gradient course of two eluent phases (eluent I: 1 M $CH_3COONa$/25 mM NaOH, eluent II: 1 M NaOH) that increased to 20% between 19 min and 20 min to separate the acidic monosaccharides. For elution condition II, an isocratic NaOH concentration of 10 mM was used, followed by the same gradient course as for elution condition I. Monosaccharide standards were obtained from Sigma-Aldrich (Germany) and included L-rhamnose (Rha), L-arabinose (Ara), D-galactose (Gal), D-glucose (Glc), D-mannose (Man), D-xylose (Xyl), D-galacturonic acid (GalUA) and D-glucuronic acid (GlcUA). Methylated monosaccharide standards were synthesized by GlycoUniverse (Germany) and included 3-O-methyl-D-xylose (3-O-Me-Xyl), 3-O-methyl-D-mannose and 6-O-methyl-D-mannose (O-methylated Man).

## Sample preparations for proteomics

Sample preparation is described briefly here, and in detail in Supplementary Note 5. Cells for all experiments were harvested at early−mid log phase ($2-4 \times 10^6$ cells/mL). Cultures for total protein extraction (quantitative experiments) were sonicated in an SDS extraction buffer followed by centrifugation to remove cell debris. Coccoliths and ESOM were isolated as described above. This material was deglycosylated with Trifluoromethanesulfonic acid (TFMS) on ice according to the manual of the GlycoProfile™ IV Chemical Deglycosylation Kit (Sigma-Aldrich). Coccolith vesicles were enriched from recalcifying cells which

had not yet exocytosed their first coccolith. After disruption using a French Press, immature coccoliths were enriched by centrifugation through a sucrose step gradient. Samples were prepared for mass spectrometry using an on-filter digest protocol followed by desalting using C18 columns. TMT labeling was performed as described in Supplementary Note 5.

## Proteomics

Samples were resuspended in 5% (v/v) acetonitrile and 0.1% (v/v) trifluoroacetic acid, before being separated on reverse phase columns (Acclaim PepMap RSLC, nanoViper: 75 μm inner diameter, 15 cm length, 2 μm bead size (Thermo Scientific)) at a flow rate of 0.4 μL min$^{-1}$ using an EASY-nLC 1000 liquid chromatograph (Thermo Fisher Scientific), which was coupled to Q Exactive HF or Q Exactive Plus mass spectrometers (Thermo Fisher Scientific) run from Xcalibur 2.1 software. An acetonitrile concentration gradient (5% to 40%) was used to elute peptides over 90 mins, or 4 h in the case of TMT-labeled samples. A final elution at 80% ACN was then performed for 1 min. Peptide ions were detected in full MS1 scan mode, with a mass-to-charge ratio range of 300–1600 and a mass resolving power of 70,000. Data-dependent tandem mass spectrometry scans were performed at a resolution of 17,500. Peptides for which MS/MS spectra had been recorded were excluded from further MS/MS scans for 30 s.

## Proteomics data analysis

The database used for all searches is available via figshare (https://doi.org/10.6084/m9.figshare.20464254.v2). It contains the predicted proteome (as described), enzymes used for digestion and common contaminants. It also included the proteomes encoded by the *E. huxleyi* chloroplast (GenBank AY741371)[84] and mitochondria (Genbank AY342361)[85]. We also included ORFs encoded by Isoseq reads that could not be mapped to the new genome which were largely derived from *Marinobacter* sequences. We gave names to these ORFs starting with the letters EhUM. Databases were concatenated with a reverse decoy database for FDR control. For analysis of the COPROs, CV, and coccosphere datasets, the Thermo raw files were converted to mzML with msconvert. Searches were carried out in Mascot[86] (2 missed cleavages, fixed mods: C carbamylation; variable modifications: M oxidation, S/T phosphorylation, precursor tolerance 10 ppm, MS2 tolerance 0.8 Da, allowed mis-cleavages 2) and with Comet[87] (peptide mass tolerance 10 ppm, fixed mods: C carbamylation; variable modifications: M oxidation, S/T phosphorylation, fragment_bin_tol 1.0005, fragment_bin_offset 0.4, allowed mis-cleavages 2). The enzyme was specified as appropriate. Mascot dat files were converted to pep.XML format using Mascot2XML from the TPP suit (TPP v5.2.0). The results for each search engine and digest enzyme (in the case of COPROs) were combined using InterProphetParser (with non-parametric modeling for FDR control) and protein inference performed with Protein-Prophet (TPP v5.2.0)[88,89]. Results were reported at a peptide probability of 0.95 and a protein probability of 0.99 (Supplementary Note 8).

The quantitative proteomics experiments (recalcification, CvsN, LvsD) involving TMT-labeling were analyzed using the Philosopher (v4.0.0)/TMT-integrator framework[90]. This uses the MS-fragger search engine[91] with full parameters available with the data in the PRIDE repository (PXD027567). TMT channels are indicated in annotation.txt files with the relevant PRIDE submissions. TMT integrator was run using an artificial reference channel and the 'Abundance_MD' normalization option. MDS plots and the correlation of replicates indicated that the normalization procedures were performing well (Supplementary Figs. 19 and 20). Differential expression between conditions was assessed using LIMMA[92], with a significance level of 0.05.

## Biochemical, spectroscopy and imaging methods

For details on SDS-PAGE, calcium quantification by ICP-OES and electron microscopy see Supplementary Note 6.

## Reporting summary

Further information on research design is available in the Nature Portfolio Reporting Summary linked to this article.

## Data availability

The raw proteomics data in the Proteomics Identification Database (PRIDE)[93] under accession codes PXD027059, PXD027440, PXD027481, PXD027501, PXD027515 and PXD027567 [https://www.ebi.ac.uk/pride/archive]. See Supplementary Table 7 for details. The raw PacBio sequencing data generated in this study have been deposited in the NCBI BioProject database under accession number PRJNA789191. The annotated genome, transcriptome and proteome level data files generated in this study have been deposited in the figshare database under accession code https://doi.org/10.6084/m9.figshare.20464254.v2. The JGI Emihu1 best proteins database used in this study is available in the JGI PhycoCosm database under the accession code Emihu. The *Arabidopsis* raw proteomics data used in this study are available in the Proteomics Identification Database (PRIDE) under accession codes PXD010580 and PXD010730. The *Arabidopsis* proteome data used in this study are available in The Arabidopsis Information Resource (TAIR) database under accession code Araport11 [https://www.arabidopsis.org/download_files/Proteins/Araport11_protein_lists/Araport11_pep_20220914.gz]. The PFAM A database [V32] used in this study is available at http://ftp.ebi.ac.uk/pub/databases/Pfam/releases/Pfam32.0/Pfam-A.hmm.gz. The EukProt protein set data used in this study for phylostratigraphic reconstruction are available in the figshare database under accession code https://doi.org/10.6084/m9.figshare.12417881.v3 [https://doi.org/10.6084/m9.figshare.12417881.v3]. Source data are provided as Source Data files. Source data are provided with this paper.

## Code availability

The code used to define the final ORF set that constitutes the Emihu2 predicted proteome is available at figshare (https://doi.org/10.6084/m9.figshare.20464293.v1).

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

## Acknowledgements

This research was supported by the Max-Planck Society and Deutsche Forschungsgemeinschaft (DFG) grant Sche1637/3-1, Sche1637/5-1 to A.Sc. A.Sc. acknowledges support from the Boehringer Ingelheim Foundation Plus3 program. A.Sk. was supported by an Alexander von

Humboldt postdoctoral fellowship. We like to thank Prof. N. Kröger for carefully reading the manuscript.

## Author contributions

A.Sk. and A.Sc. designed the research; A.F., A.Sk., C.W., and B.H. contributed the improved genome and transcriptome; M.B. contributed the COPRO data; S.S. contributed the coccolith vesicle data; A.Sk. contributed the recalcification CvsN, LvsD and coccosphere data, and analyzed the data of all proteomics experiments; M.G. and A.G. contributed to all proteomics experiments; A.Sc. contributed the CAPs data; A.Sk. and A.Sc. wrote the paper with the input of all co-authors.

## Funding

## Competing interests

The authors declare no competing interests.
