## [Peer Review File · Nature Communications]

A joint proteomic and genomic investigation provides insights into the mechanism of calcification in coccolithophoresREVIEWER COMMENTS

Reviewer #1 (Remarks to the Author):

In this paper, Skeffington and coworkers, by using the single celled calcifying algae *Emiliania huxleyi* as a model organism, have identified the molecular machinery that is required for the biosynthesis of coccoliths. To this end, they have used a complete - and impressive - arsenal of high throughput techniques including genomics, transcriptomics and proteomics. To date, this work represents probably the most comprehensive study on the calcifying proteome of *E. huxleyi*. In particular, it appears that the data presented here represent a much more complete set than what had been published before in the *Emilhu1* proteome. One of the strengths of the paper is the use of a "multifaceted strategy" (as announced in the title) to identify the numerous proteins involved in different manners (ion transporters, cytoskeletal proteins, enzymes, proteins involved in sugar metabolism...) in the intravesicular calcification of the coccoliths.

By doing so, the authors have obtained counter-intuitive finding: the combination of the proteomic and polysaccharide data, along with the speed of the recalcification response, support the hypothesis that the calcification machinery continues to be produced under low calcium conditions, despite an inability to produce calcite due to insufficient Ca^{2+} availability. In other words, the availability of extracellular calcium is not an activator of the genetic program underlying coccolith formation. Among other interesting findings on the calcifying machinery, one cites the importance of cytoskeletal proteins and the abundance of proteins of unknown function, two facts that are also observed in other calcifying organisms.

I have only few remarks / questions about this well-written paper:

- All figures are very informative, but a little bit "crowded" and difficult to read. Is there a way to lighten them?

- In the "Phylostratigraphic analysis" § (page 6 and Fig. 2), we understand that a major proportion of orthogroups (4941) are *E. huxleyi* specific, suggesting that the calcifying machinery for making coccoliths is a fast evolving system. Can however the authors provide an evaluation of the percentages of de novo genes that were recruited at different stages (Haptophyta, Calcihaptophyta, Isochrysidales) of the phylogenetic tree of Fig. 2? Is there a way to put dates at the different branching of the tree of figure 2?

- In the discussion and following my remark above, a short paragraph about evolution would have been welcome. How were these novel genes implemented in the *E. huxleyi* clade? By gene duplication? Exon shuffling? Or other mechanisms?

- *E. huxleyi*, by its calcifying activity, plays a key-role in climate homeostasis (: a sink for CO_2). In the discussion, a couple of sentences (or a short §) linking the importance of understanding the calcifying machinery of *E. hux.* to climatic issues would be welcome.

I have some minor remarks, listed hereunder. Some are about typos or misspelling, some others concern very specific questions.

Abstract:

Line 20: Working on rather than working in.

Line 22: These analyses are underpinned by a new genome = are supported (to avoid repeating "underpin").

Main text:

Lines 47-48: "it may be possible to use this knowledge for advances in nanotechnology". Can the authors give a couple of examples (very shortly) of the potential use of the knowledge for advances in nanotechnology instead of simply giving a reference?

Line 116: *Arabidopsis*: in italic (see also lines 576, 583, 585).

Line 152: *Gephyrocapsa*: in italic.

Line 171: recalcification = recalcification.

Line 198: Stains-All (with an 's').

Line 213, legend Fig. 3: SDS-PAGE.

In figure 3, there is no glucose in the composition of the polysaccharides? What about glucosamine and galactosamine?

Line 219: embedded or occluded?

Line 225: how can the authors be sure that the smear is not due to degradation caused by the

chemical deglycosylation? Was the chemical deglycosylation performed at 0°C?
Line 250 and following: (not mandatory) have the authors tried to run 2D gels to 'deconvolute' (so to speak) the thick 40 kDa-band observed in 1D gels?
Figure 4, gel b: the values for the low molecular weight standards (10 and 15 kDa) are shifted.
Line 268: cells grown in low calcium medium...
Line 294: we identified several proteins of unknown function, with high predicted intrinsic disorder: how many? Percentages?
Line 303: Proto-coccolith calcite is protected from dissolution by EDTA in the absence but not presence of SDS = unclear, what does it mean?
Line 304: SDS-PAGE.
Line 339: *Gephyrocapsa*-age = *Gephyrocapsa* in italic.
Line 381: `:' (after peptide repeats) should be replaced by a comma.
Line 390: HCO₃⁻ instead of HCO₃⁻.
Line 594: *E. huxleyi*: in italic.
Line 640: monosaccharides.
Line 659: was the TFMSA deglycosylation performed at 0°C or at room temperature? Please, indicate the temperature of deglycosylation.
Line 685: *Marinobacter* in italic.
Line 686: These ORFs we gave names starting with the letters EhUM. Change the sentence: we gave names to these ORFs...
Line 699: recalification = recalcification.

Reference list:

Here and there (ref. 9, 12, 14, 15, 18...) in article's titles, put species names in italic.

In conclusion, this interesting and original paper is a major and noteworthy contribution to the field of biomineralization. The methodology employed by the authors is sound and impressive and the conclusions are well supported by the data. I therefore recommend its publication by Nature Comm. after the required corrections.

Reviewer #4 (Remarks to the Author):

Summary and major comments

This study represents a bold attempt to advance the identification of the molecular mechanisms of calcification in the model marine coccolithophore *E. huxleyi*. Although these organisms are the most significant producers of biogenic calcite in the oceans, our ability to model how they influence biogeochemical cycles in the past, present, and future oceans is hampered by our limited understanding of the calcification process itself. Moreover, a more thorough understanding of coccolithophore calcification is likely to yield important insights that are highly relevant to biotechnology and nanoscience. Despite decades of research and the availability of a variety of sequencing resources, several barriers have prevented the identification of the calcification machinery with confidence. Here, the authors use long-read sequencing approaches to convincingly establish a significantly improved *E. huxleyi* genome and a high-quality predicted proteome. The authors then carried out a suite of proteomic analyses on coccoliths, extracellular matrix, and isolated coccolith vesicles to identify proteins most likely involved in the calcification process. The authors should be commended for this extensive work which no doubt required a great deal of persistence and technical troubleshooting to accomplish. The study represents a significant step forward for the community. While there are some interesting new insights, the main weakness is that ultimately, after all these elegant experiments, our current (quite limited) molecular model for calcification-related genes and proteins remains little changed (e.g. Figure 7), with the functional roles of a few 'new players' remaining equivocal or overly speculative. This should not necessarily be a bar to publication here or elsewhere, but it is reasonable to point out.

Note: I do not feel qualified to critically evaluate the details of the technology, methodology and software tools used for the sequencing and proteomics in this study.

Detailed comments:

Overall, the manuscript is very well written and clear. The experimental approaches are novel and the data generally appear robust. Here are comments/corrections the authors should address:

Line 32 Is the citation '3a' correct?

Line 123 Do they mean 'far higher'?

Lines 153-157 and Figure 2, Please be clear on how these overrepresented Pfam groups were selected/identified. Was there some cut off or threshold used? Or was this selection based on prior knowledge/expectation? Are there any other Pfam groups not included here that showed a similar progression or loss or enrichment that might be informative.

Figure 2 and associated discussion. There is a missed opportunity to evaluate and discuss the Isochrysidales, which appears to follow the progression of Pfam enrichment over time in the calcihaptophyta yet this group has lost the ability to calcify! The analogy here is the authors' observations that the expression of the calcification machinery in low [Ca] treatments remains intact even though calcification is inhibited. I think this would be an important and interesting point to discuss.

Section starting Line 169.

Here and Figure 3 legend please specify how many generations/time cultures were maintained in low Ca²⁺ prior to the recalcification response experiment. The result is not counterintuitive if the recalcification occurs only after a short period in low Ca²⁺. This detail in the text will help avoid confusion.

Line 200-202 This is an interesting and important insight; the cellular pools of Ca²⁺ are insufficient to supply calcification!

Line 221 Typo: 'Followed' not following

Line 229. What were the proteins/functions and criteria used to exclude them? Might others not also be contaminants esp. 'novel' proteins and not be directly associated with calcification?

Line 234. I think the argument that CV proteins might be expected to have ER signal sequences so they can be transported into the Golgi system is wrong. The CV and baseplate scales that template calcification are derived from the Golgi system with nucleation and maturation of the calcite coccolith essentially occurring 'trans-Golgi'. Therefore the CV is the result of export from the Golgi, not to it.

Line 239. "the most trivial": I think this has been a major limitation and suggest rephrasing.

Line 275. While the authors provide evidence that they successfully isolated protococcoliths bound by CV membrane, it is important to acknowledge the close association of ER and reticular body membranes that have been demonstrated through numerous TEM ultrastructure studies. Thus, the surrounding membrane(s) of isolated CVs are not likely comprised of just a single lipid vesicle membrane. There will be numerous 'extra-vesicular' materials. Indeed, the authors themselves find evidence of actin and myosin in these samples (see also line 385), highlighting the tight association of the CV membrane with the extra-vesicular environment, including adjacent ER-like membranes. Could the V-type ATPase signal in fact be more associated with the adjacent ER and not the CV itself?

Related to the above remarks, interpretation of the SEM done to visualize the CV membrane of isolated protococcoliths (Supplementary Figure 9) needs to be revised or excluded. The authors interpret the different contrast of backscattered electron images (Panel B) to indicate CV membrane (low signal/darker greyscale) compared to the calcite (higher signal/brighter greyscale). The areas interpreted as membrane are almost certainly the organic baseplate scale and not the very thin lipid bilayer of the vesicle if it was present. The escape volume of BSE is much deeper than SE and unlikely to give sufficient signal/contrast from just the CV membrane, if present, using the column conditions stated. Also classical freeze fracture typically allows for low energy secondary electron imaging from the cleaved lipid membrane surfaces. I do not see typical evidence one could expect indicating the presence of a membrane in the panel A image. The

supplementary figure 9 is not sufficiently compelling in my view. Cryo-Electron tomography with TEM would be the best approach to resolve the architecture of the membranes and associated material of isolated protococcoliths.

Line 331 -334. The authors suggest their analysis of coccosphere proteins may be compromised by the production of extracellular vesicles. Please discuss how this dynamic exchange of plasma-membrane derived materials from cell to extracellular space affects your interpretation. These vesicles, which may have little to do with calcification per se could be the origin of several of the proteins considered to be calcification-specific.

Line 392- This is a significant finding

Line 380. This is very intriguing!

Line 416. I found the supplementary note 7 tables confusing/unclear to read. Did the authors actually do PCR using the Mackinder et al. primer sets on each of these samples, or are all the ticks and crosses simply indicating in-silico predictions of whether the candidate genes would be amplified? Some additional explanation and guidance on how to read the tables would be helpful.

Lines 424-433 this section is too speculative. For example, how could the SLC24 transporter be involved in regulating Ca^{2+} in the coccosphere (extracellular) microenvironment when seawater Ca^{2+} is ~ 10 mM? Please clarify the rationale here. Another example; if V-type ATPase proteins are up-regulated in N-cells vs C-cells then, using previous criteria, this should equally be interpreted evidence against a role in calcification correct? Also, because production of calcite from bicarbonate in the CV produces H^+ , over-acidification of the CV will happen when H^+ removal is compromised, not necessarily because of over activation of a V-type ATPase.

Line 455- A helpful advance/resolution of a long-standing question

Line 466 -End of sentence does not make grammatical sense. Please revise.

Line 594 Italics needed for species

RESPONSE TO REVIEWERS' COMMENTS

Reviewer #1 (Remarks to the Author):

In this paper, Skeffington and coworkers, by using the single celled calcifying algae *Emiliana huxleyi* as a model organism, have identified the molecular machinery that is required for the biosynthesis of coccoliths. To this end, they have used a complete - and impressive - arsenal of high throughput techniques including genomics, transcriptomics and proteomics. To date, this work represents probably the most comprehensive study on the calcifying proteome of *E. huxleyi*. In particular, it appears that the data presented here represent a much more complete set than what had been published before in the Emilhu1 proteome. One of the strengths of the paper is the use of a “multifaceted strategy” (as announced in the title) to identify the numerous proteins involved in different manners (ion transporters, cytoskeletal proteins, enzymes, proteins involved in sugar metabolism...) in the intravesicular calcification of the coccoliths.

By doing so, the authors have obtained counter-intuitive finding: the combination of the proteomic and polysaccharide data, along with the speed of the recalcification response, support the hypothesis that the calcification machinery continues to be produced under low calcium conditions, despite an inability to produce calcite due to insufficient Ca^{2+} availability. In other words, the availability of extracellular calcium is not an activator of the genetic program underlying coccolith formation. Among other interesting findings on the calcifying machinery, one cites the importance of cytoskeletal proteins and the abundance of proteins of unknown function, two facts that are also observed in other calcifying organisms.

I have only few remarks / questions about this well-written paper:

- All figures are very informative, but a little bit "crowded" and difficult to read. Is there a way to lighten them?

We have critically evaluated all figures and identified a few sub-panels that are not of crucial importance for understanding the content of the paper and following our conclusions. These sub-panels we have removed from figures of the main text and added as individual figures to the supplementary material. One sub-panel has been converted into a new main figure. These

changes allowed us to enlarge the remaining sub-panels in the figures, resulting in better readability. In detail, the following changes were made:

- The panel “key summary statistics of Emihu2 compared to Emihu1”, formerly Fig. 1a, is now included as Table 1.
- The panel “Experimental design for generation of *E. huxleyi* long-read transcriptome”, formerly Fig. 1b, is now Supplementary Fig. 1.
- The panel “Experimental design for the isolation of coccoliths and soluble coccolith-associated organic material (SCAOM) for biochemical and proteomic analysis.” panel, formerly Fig. 4a, is now Supplementary Fig. 7.
- The panel “Experimental setup for the identification of coccosphere associated proteins.” panel, formerly Fig. 6a, is now Supplementary Fig. 12.
- The panel “Proposed molecular organization of the CV and the coccosphere” panel, formerly Fig. 7b, is now included as individual figure (Fig. 8).

- In the "Phylostratigraphic analysis" (page 6 and Fig. 2), we understand that a major proportion of orthogroups (4941) are *E. huxleyi* specific, suggesting that the calcifying machinery for making coccoliths is a fast evolving system. Can however the authors provide an evaluation of the percentages of de novo genes that were recruited at different stages (Haptophyta, Calcihaptophyta, Isochrysidales) of the phylogenetic tree of Fig. 2? Is there a way to put dates at the different branching of the tree of figure 2?

- In the discussion and following my remark above, a short paragraph about evolution would have been welcome. How were these novel genes implemented in the *E. huxleyi* clade? By gene duplication? Exon shuffling? Or other mechanisms?

The reviewer raises some very interesting points, which we are happy to address.

First, we would like to bring to mind again that the purpose of the phylostratigraphic analysis was to prioritize the proteomically identified proteins for future studies based on correlating their evolution with the emergence of calcification in the phylogeny. It served this purpose well, and although further detail about the mechanisms of gene emergence and evolution would doubtless be fascinating, we do not consider it necessary to understand these mechanisms in order to use the phylostratigraphic analysis for candidate prioritisation.

To comment on the authors accurate observation that many orthogroups appear to be *E. huxleyi* specific. We think this is very likely the case, but the conclusion must be tempered with the caveat that no long-read transcriptomic data of equivalent depth has been used to identify genes in any other the other sequenced haptophyte species. Thus the high quality of our data could potentially result in orthogroups appearing *E. huxleyi* specific, although they are in reality more widespread. This point is already made in the text, lines 147 – 148. Even taking this caveat into consideration, we would argue that the conclusion that there has been considerable genetic innovation in *E. huxleyi* is reasonable, since any orthogroup widespread among the haptophytes and expressed to moderate levels would likely have been detected at least once in the extensive multi-species transcriptome sequence data used as part of the phylostratigraphic analysis.

To address the reviewers query about de novo genes, we would like to make sure we all have the same understanding of what "de novo genes" are. For us, they are genes arising when non-coding DNA acquires transcriptional activity and genetic changes that result in a functional product. De novo genes are very difficult to identify. In theory, they might be identifiable by looking for orthogroups which lack homology to genes that are older than the oldest phylostratigraphic level in which the orthogroup is detected. In practice, the search for genes of de novo origin suffers from the fact that anciently diverged homologues can be very difficult to detect so that a very high false positive rate is to be expected. In addition to divergence beyond recognition, potential composite origins of genes and the need for excellent genomic data across the entire phylogeny of interest - the current bottleneck for comparative genomics of algae - further limit the identification of de novo genes. For these reasons, the only convincing demonstrations of de novo gene origins to date have come from in-depth analyses of 'orphan' genes, found only in a single species. The most robust approach for de novo gene identification relies on syntenic relationships (DOI: [10.1371/journal.pgen.1008160](https://doi.org/10.1371/journal.pgen.1008160)), an option that is currently not open to use given that many of the relevant species in our study have only transcriptome data available and if genomes are available, they are often fragmented. For these reasons, we are reluctant to report percentages of de novo genes recruited at different evolutionary stages.

The reviewer also raises the broader question of how the genes exclusive to *E. huxleyi* have evolved and whether this involved processes such as gene duplication or exon shuffling. Given that these orthologues have only been identified within the one species, answering this question requires investigations into evolution of the sequences at the sub-gene, domain or sequence pattern level. A few pipelines have been developed for these analyses (DOI: [10.1093/gbe/evx069](https://doi.org/10.1093/gbe/evx069), DOI: [10.1093/molbev/msr222](https://doi.org/10.1093/molbev/msr222)). Using them requires substantial time investment to develop the multi-genome de novo domain databases required as input. We would argue that investigating the mechanisms of *E. huxleyi* specific gene evolution would not improve our ability to identify the calcification machinery sufficiently for the investment required, so we consider such investigations out of the scope of this paper. Indeed, we feel that our current phylostratigraphic analysis well serves its main purpose even without knowing the mechanism by which the identified genes have evolved.

However, without further complex analyses, we can make some statements about the origins of *E. huxleyi* genes that might be informative:

1. Genes exclusive to *E. huxleyi* that are part of the same "orthogroup" must be paralogues of each other – i.e. have arisen from duplication of an ancestral *E. huxleyi* gene. 10,939 genes fall into this category, or 13% of *E. huxleyi* loci (see table Emihu2_proteome_annotation.txt at doi.org/10.6084/m9.figshare.20464254.v1)
2. 46% of *E. huxleyi* specific genes have domain annotations (assessed with pfam, KEGG and EggNog mapper, see table Emihu2_proteome_annotation.txt at doi.org/10.6084/m9.figshare.20464254.v1). In these cases, some portion of the gene certainly has an origin older than *E. huxleyi*, suggesting that processes such as fusion of genic elements, or gene duplication combined with divergence outside the known domain, could be at play.

We have modified the text to include some discussion of the points made above:

A large number of orthogroups were *E. huxleyi* specific (4,941, Fig. 2), which may to some degree reflect the fact that similarly detailed genomic and transcriptomic information are not available for other haptophytes, while 724 orthogroups appeared to be specific to the Calcihaptophycidae clade containing all coccolithophores.

has been changed to:

Many orthogroups were *E. huxleyi* specific (4,941, Fig. 2), which may to some extent reflect the fact that similarly detailed genomic and transcriptomic information are not available for other haptophytes. However, it likely also reflects genuine innovation in the *E. huxleyi* clade, as we would expect orthogroups widespread among the haptophytes would have been detected at least once in the transcriptome and genome data from multiple species used in the phylostratigraphic analysis. These recently evolved genes represent 13% of *E. huxleyi* genes, although 46% of those contain known domains, implying more ancient DNA sequence was often involved in their origins, perhaps through processes such as domain fusion. The 724 orthogroups specific to the Calcihaptophycidae clade, which contains all coccolithophores, are of particular interest for understanding the mechanism of calcification.

Is there a way to put dates at the different branching of the tree of figure 2?

Based on the fossil record and previous research it is possible to put approximate dates at the branch points. However, for the purposes of this figure, relating branch points to other data, it is necessary that the tree is a cladogram rather than a phylogram. Only the latter has branch lengths proportional to evolutionary time. Adding dates to a cladogram could be misleading as it suggests a meaning for the y-axis scale when there is none. For this reason, we prefer to leave the figure as it is, without dates.

- *E. huxleyi*, by its calcifying activity, plays a key-role in climate homeostasis (: a sink for CO₂). In the discussion, a couple of sentences (or a short §) linking the importance of understanding the calcifying machinery of *E. hux.* to climatic issues would be welcome.

Following the suggestion of the reviewer we added the following sentences to the beginning of the discussion:

This biomineralization process has consequences for the global carbon cycle because it impacts the exchange of CO₂ between the ocean and the atmosphere and the burial of carbon on the ocean floor. It is therefore critical to know more about the biology of this process to be able to understand how coccolithophores will respond to future changes in ocean chemistry and climate in order to make more accurate predictions about the future of the carbon cycle and the impacts of climate change.

I have some minor remarks, listed hereunder. Some are about typos or misspelling, some others concern very specific questions.

Abstract:

Line 20: Working on rather than working in.

Modified

Line 22: These analyses are underpinned by a new genome = are supported (to avoid repeating “underpin”).

Modified

Main text:

Lines 47-48: “it may be possible to use this knowledge for advances in nanotechnology”. Can the authors give a couple of examples (very shortly) of the potential use of the knowledge for advances in nanotechnology instead of simply giving a reference?

To clarify how an advanced understanding of the mechanism of coccolith morphogenesis may contribute to advances in nanotechnology we have revised the corresponding sentence to:

The extraordinary morphological patterning of coccoliths at the micro- and the nano-scales means they have potential as components of novel materials²², and if we can understand the complex control of crystal morphologies that takes place in coccolithophores, it may be possible to use this knowledge for developing new methods for producing complex arrays of inorganic crystals for applications in sensing, catalysis, and photonics^{23,24}.

Line 116: Arabidopsis: in italic (see also lines 576, 583, 585).

Modified

Line 152: Gephyrocapsa: in italic.

Modified

Line 171: recalification = recalcification.

Modified

Line 198: Stains-All (with an ‘s’).

Modified

Line 213, legend Fig. 3: SDS-PAGE.

Modified

In figure 3, there is no glucose in the composition of the polysaccharides? What about glucosamine and galactosamine?

Figure 3 shows the quantification of all monosaccharides that are accepted to be constituents of the CAPs and that we could quantify. Whether glucose is a constituent of the CAPs is still unclear. Borman et al. 1987 did not report glucose as a component of the CAPs of three *Emiliana* strains. Fichtinger-Schepman et al. 1981 however reported glucose in the CAP and then in molar amounts similar to that of arabinose. We detected glucose in our CAP hydrolysate (Supplementary figure 6b, at ~9,7 min retention time). However, the area under the peak was always below that of the lowest concentrated glucose standard (25 nM) and therefore it is too small for quantification. In contrast to the results of Fichtinger-Schepman et al. the amount of glucose in our samples is tiny compared to arabinose. Our data therefore suggests that glucose is only a trace component of the CAPs of strain AWI1516. At this point we would like to bring to mind that many abundant plant and algal polysaccharides do not contain glucose, such as galactomannans, arabinoxylans, fucans, and carrageenans.

In the exudate of cells grown in low calcium and also high calcium medium, high concentrations of glucose were detected (Supplementary figure 6). From control samples without TFA hydrolyzation we know that the glucose is part of a polysaccharide (Supplementary figure 6b). We further know from microscopy analysis of our cultures that low calcium grown cells are less robust than cells grown at standard-calcium concentrations and lyse more easily. We consider it therefore likely that some cells may have lysed during preparation of the extracellular EDTA-soluble organic material, releasing glucose-rich polysaccharides such as laminarin.

For these reasons we have omitted glucose from the list of monosaccharides shown in figure 3. The methylated monosaccharides could also not be quantified.

To make the reader aware that some of the detected monosaccharides could not be quantified we added the following sentence to the legend of figure 3b:

Note that not all monosaccharides that were detected could be quantified. For chromatograms see Supplementary figure 6.

And to the legend of Supplementary figure 5 we added:

Note that glucose in the CAPs hydrolysate was below the concentration range for quantification, and that the methylated monosaccharides could not be quantified due to technical difficulties.

We did not detect any glucosamine or galactosamine in our samples, nor did previous analyses of CAPs by Borman et al. 1987 and Fichtinger-Schepman et al. 1981.

Borman, A. H. et al. Coccolith-associated polysaccharides from cells of *Emiliana huxleyi* (Haptophyceae). J Phycol 23, 118-123 (1987).

Fichtinger-Schepman, A. M., Kamerling, J. P., Versluis, C. & Vliegenthart, J. F. G. Structural studies of the methylated, acidic polysaccharide associated with coccoliths of *Emiliana huxleyi* (Lohmann) Kämtner. Carbohydrate Research 93, 105-123 (1981).

Line 219: embedded or occluded?

Modified to occluded.

Line 225: how can the authors be sure that the smear is not due to degradation caused by the chemical deglycosylation? Was the chemical deglycosylation performed at 0°C?

The reviewer hits on a tricky point of chemical deglycosylation, namely that not only polysaccharides but also proteins can get degraded. For that reason, we cannot exclude the possibility that the smear in the SDS-PAGE represents degraded proteins. It may be degradation products of the 40 kDa protein and/or other proteins. In the latter case, these proteins must be very sensitive to the deglycosylation procedure, since SDS-PAGE analyses of more than 30 deglycosylation experiments that we performed with different batches of isolated material never revealed additional protein bands to those visible in Fig. 4a. Finally, we want to bring to mind again, that we only draw conclusion from the remaining. However, we like to point out that we only

We have modified the respective sentence, which now reads:

SDS-PAGE and silver staining of the deglycosylated material revealed a prominent band at ~40 kDa, which was protease sensitive and therefore represents a protein, as well as a smear across most molecular weights that likely represents degradation products (Fig. 4a).

The deglycosylation reaction was carried out on ice according to the manual provided with the kit. We have added this information to the Materials and Methods.

Line 250 and following: (not mandatory) have the authors tried to run 2D gels to 'deconvolute' (so to speak) the thick 40 kDa-band observed in 1D gels?

No, we did not run 2D gels.

Figure 4, gel b: the values for the low molecular weight standards (10 and 15 kDa) are shifted.

Fixed.

Line 268: cells grown in low calcium medium...

Modified.

Line 294: we identified several proteins of unknown function, with high predicted intrinsic disorder: how many? Percentages?

We have added the number of identified proteins to the corresponding sentence. It now reads:

In addition, we identified eight proteins of unknown function, with high predicted intrinsic disorder, and low sequence disorder (Fig. 5e).

Line 303: Proto-coccolith calcite is protected from dissolution by EDTA in the absence but not presence of SDS = unclear, what does it mean?

We have modified the respective sentence to:

Isolated proto-coccolith calcite is protected from dissolution by the calcium chelator EDTA. After addition of SDS detergent, which solubilizes membranes, proto-coccolith calcite is dissolved by EDTA.

Line 304: SDS-PAGE.

We have used SDS-PAG as acronym for sodium dodecylsulfate polyacrylamide gel. The picture shows a gel and not the sodium dodecylsulfate polyacrylamide gel electrophoresis (SDS-PAGE). Since SDS-PAG does not seem to be a very commonly used abbreviation, we have modified "SDS-PAG" to "SDS-PAGE gel".

Line 339: Gephyrocapsa-age = Gephyrocapsa in italic.

Modified.

Line 381: ':' (after peptide repeats) should be replaced by a comma.

Modified.

Line 390: HCO₃⁻ instead of HCO₃⁻.

Modified.

Line 594: *E. huxleyi*: in italic.

Modified.

Line 640: monosaccharides.

Modified.

Line 659: was the TFMSA deglycosylation performed at 0°C or at room temperature?
Please, indicate the temperature of deglycosylation.

We have used the GlycoProfile™ IV Chemical Deglycosylation Kit from Sigma-Aldrich for deglycosylation and followed the instructions given in the manual. There it is suggested to perform the reaction on ice or at 2 - 8°C. We deglycosylated the samples on ice. We have added this information to the Materials and Method section.

Line 685: *Marinobacter* in italic.

Modified.

Line 686: These ORFs we gave names starting with the letters EhUM. Change the sentence: we gave names to these ORFs...

Modified.

Line 699: recalification = recalcification.

Modified.

Reference list:

Here and there (ref. 9, 12, 14, 15, 18...) in article's titles, put species names in italic.

Modified.

In conclusion, this interesting and original paper is a major and noteworthy contribution to the field of biomineralization. The methodology employed by the authors is sound and impressive and the conclusions are well supported by the data. I therefore recommend its publication by Nature Comm. after the required corrections.

Reviewer #4 (Remarks to the Author):

Summary and major comments

This study represents a bold attempt to advance the identification of the molecular mechanisms of calcification in the model marine coccolithophore *E. huxleyi*. Although these organisms are the most significant producers of biogenic calcite in the oceans, our ability to model how they influence biogeochemical cycles in the past, present, and future oceans is hampered by our limited understanding of the calcification process itself. Moreover, a more thorough understanding of coccolithophore calcification is likely to yield important insights that

are highly relevant to biotechnology and nanoscience. Despite decades of research and the availability of a variety of sequencing resources, several barriers have prevented the identification of the calcification machinery with confidence. Here, the authors use long-read sequencing approaches to convincingly establish a significantly improved *E. huxleyi* genome and a high-quality predicted proteome. The authors then carried out a suite of proteomic analyses on coccoliths, extracellular matrix, and isolated coccolith vesicles to identify proteins most likely involved in the calcification process. The authors should be commended for this extensive work which no doubt required a great deal of persistence and technical troubleshooting to accomplish. The study represents a significant step forward for the community. While there are some interesting new insights, the main weakness is that ultimately, after all these elegant experiments, our current (quite limited) molecular model for calcification-related genes and proteins remains little changed (e.g. Figure 7), with the functional roles of a few 'new players' remaining equivocal or overly speculative. This should not necessarily be a bar to publication here or elsewhere, but it is reasonable to point out.

Note: I do not feel qualified to critically evaluate the details of the technology, methodology and software tools used for the sequencing and proteomics in this study.

Detailed comments:

Overall, the manuscript is very well written and clear. The experimental approaches are novel and the data generally appear robust. Here are comments/corrections the authors should address:

Line 32 Is the citation '3a' correct?

It seems that this reference was not included in the manuscript. We have corrected this. The added reference, now number 3, is:

Segev, E. *et al.* Dynamic metabolic exchange governs a marine algal-bacterial interaction. *Elife* **5**, e17473 (2016).

Line 123 Do they mean 'far higher'?

Modified.

Lines 153-157 and Figure 2, Please be clear on how these overrepresented Pfam groups were selected/identified. Was there some cut off or threshold used? Or was this selection based on prior knowledge/expectation? Are there any other Pfam groups not included here that showed a similar progression or loss or enrichment that might be informative.

The statistical method used is explained in the method section under "Phylostratigraphic analysis". There we wrote: "Pfam domains were identified using InterProScan (InterPro 82.0). Enriched domains were identified by calculating hypergeometric p-values from which q-values were estimated using the R function `qvalue_trunc.R`. A q-value estimate < 0.05 was required for a domain to be considered enriched."

For clarity, we have added the false discovery rate threshold in the main text (line 162 - 164):

We assessed which Pfam domains were over-represented in each orthogroups-age class relative to the entirety of the *E. huxleyi* proteome (qvalue < 0.05, Supplementary Data 6), then grouped those domains by biological role (Fig. 2).

The groups were chosen for display because they are theoretically of interest for calcification, and because they generally display more enrichment in calcifying lineages. Note that progressive enrichment (mentioned by the reviewer) through the phylogeny towards *E. huxleyi* is not necessarily a characteristic that would make us link a Pfam domain to calcification. For example, there are many coccolithophores outside of the Gephyrocapsa clade which are more heavily calcified than *E. huxleyi*. Many enriched Pfam domains outside the Pfam groups highlighted in Fig. 1 are difficult to interpret in the context of calcification or may relate to other physiological adaptations. However, we accept that readers may want to easily explore this data for themselves, so we have now included the Pfam enrichment table for the haptophyte part of the tree as Supplementary Data 6, which we refer in the main text at line 164.

Figure 2 and associated discussion. There is a missed opportunity to evaluate and discuss the Isochrysidales, which appears to follow the progression of Pfam enrichment over time in the calcihaptophyta yet this group has lost the ability to calcify! The analogy here is the authors' observations that the expression of the calcification machinery in low [Ca] treatments remains intact even though calcification is inhibited. I think this would be an important and interesting point to discuss.

We thank the reviewer for pointing this out, and agree that an examination of the proteins lost in the Isochrysis genus might be informative for identifying calcification related proteins. We have added the following sentence to the text in the section relating to the phylostratigraphic analysis:

Line 160:

Intriguingly, 2,156 orthogroups of haptophyte age or younger have been lost in non-calcifying Isochrysis, but are retained in *E. huxleyi*, and 254 of these are of Calcihaptophycidae age.

In addition, we have referenced this protein set when exploring the COPRO dataset:

Line 271-273 (end of paragraph):

It was striking that six COPROs (EhG13787.3, EhG17242.1, EhG17242.7, EhG21037.1, EhG21537.1 and EhG30161.1) were in the set of 254 proteins of Calcihaptophycidae age lost in non-calcifying *Isochrysis*.

We have updated Supplementary Data 2 (Table of COPROs) to include a column specifying whether or not the orthogroup has been lost in Isochrysis:

Finally, we have updated Supplementary Table 5 to highlight a protein (EhG4644.3) showing overlap between multiple datasets and also lost in Isochrysis:

When answering this point we also noticed that a label in fig. 2 is inaccurate: The x-axis label 'Isochrysidales' should in fact read Isochrysidaceae, indicating the family of non-calcifying Isochrysidales (the order). We have revised the label in the new version of fig. 2.

For readers interested in performing more in-depth analyses with the phylostratigraphic data, the complete analysis is downloadable from Figshare, as described on line 621 (doi.org/10.6084/m9.figshare.20463900).

Section starting Line 169.

Here and Figure 3 legend please specify how many generations/time cultures were maintained in low Ca²⁺ prior to the recalcification response experiment. The result is not counterintuitive if the recalcification occurs only after a short period in low Ca²⁺. This detail in the text will help avoid confusion.

The minimum time that the cultures were grown in low-Ca medium before being used in an experiment is given in the materials and methods section. It is 2 weeks. Extracellular polysaccharides were isolated from cultures that were 20 days old, which is displayed in figure 3a. **The low-Ca culture for polysaccharide isolation was started with cells that had been propagated in low-Ca medium for 1.5 month. The recalcification experiment was performed with cells that had been in low-Ca medium for 14 days.**

We added these two sentences to the legend of figure 3.

Line 200-202 This is an interesting and important insight; the cellular pools of Ca²⁺ are insufficient to supply calcification!

We prefer to keep our conclusion as it is without bringing potential cellular pools of calcium into play. We have not checked the cells from these experiments for the presence of calcium storage pools. From previous work, where the cells were grown for more than 6 months in low-Ca medium [Gal et al. 2017], we know that these cells lack the calcium pools that have been observed in cells grown in Std-Ca medium and are referred to as Ca-bodies in the literature. Therefore we hypothesize that the cells used in the recalcification experiment also lack Ca-bodies.

Gal, A. et al. Trace-element incorporation into intracellular pools uncovers calcium-pathways in a coccolithophore. *Advanced Science* 4, 1700088 (2017).

Line 221 Typo: 'Followed' not following

Modified.

Line 229. What were the proteins/functions and criteria used to exclude them? Might others not also be contaminants esp. 'novel' proteins and not be directly associated with calcification?

We classified proteins with strong homology to proteins with known function in evolutionary conserved processes as contaminant proteins, such as histones, enzymes of major primary metabolic pathways and photosynthesis proteins. It's very difficult to completely get rid of some of these generally highly abundant proteins during coccolith preparations, meaning that a few residual peptides are detected. The reviewer is correct that a proportion of the other COPROs identified could also be proteins not directly related to calcification. We have used several strategies to mitigate for this, including focussing on proteins containing a signal peptide (see below), focussing on the protein set that overlaps with other datasets described in the paper and focussing on proteins that arose in calcifying lineages of the haptophytes. We have no reason to suspect any protein in our final list of COPROs of being a contaminant, in contrast to the excluded proteins, where good evidence exists that these are likely contaminants.

Line 234. I think the argument that CV proteins might be expected to have ER signal sequences so they can be transported into the Golgi system is wrong. The CV and baseplate scales that template calcification are derived from the Golgi system with nucleation and maturation of the calcite coccolith essentially occurring 'trans-Golgi'. Therefore the CV is the result of export from the Golgi, not to it.

We are not sure we entirely understand the reviewer's comment here. The baseplate certainly develops in the trans-Golgi, and we would define a vesicle containing a base-plate an early stage coccolith vesicle. We are not aware of mechanisms for a protein to reach the trans-Golgi without trafficking via the ER and cis/medial Golgi. Proteins can be targeted to the TGN via endocytosis, which could conceivably be part of the system recycling components to a new coccolith vesicle after exocytosis of a coccolith, but even these proteins would require initial export via the secretory system. Thus, we consider it entirely reasonable to prioritise proteins with signal peptides.

To clarify the most likely route for deliver of proteins into the CV we have modified the sentence addressing this aspect to:

The mechanisms of protein targeting to coccolith vesicles are unknown, but given that there is ultrastructural evidence for the coccolithophore genus *Pleurochrysis* that CVs originate from the Golgi, it seems likely that luminal CV proteins should contain a signal peptide for import into the endoplasmic reticulum, from where they are transported to the Golgi system and from there into the CV.

Line 239. "the most trivial": I think this has been a major limitation and suggest rephrasing.

We accept the reviewers point that our knowledge of endomembrane targeting is far from complete, and that this limits our ability to make predictions about localization from sequence alone. It is widely accepted that proteins can enter the secretory system independently of the SRP or Sec61 pathways for example, although the mechanistic basis of this is still poorly understood (DOIs: [10.1038/nature20169](https://doi.org/10.1038/nature20169), [10.1016/j.cell.2013.02.003](https://doi.org/10.1016/j.cell.2013.02.003)). Given that we know our COPRO dataset contains some contaminants (e.g. some highly abundant metabolic proteins),

we prefer to be conservative and focus on those proteins in the data set containing a known secretory targeting sequence. We consider it possible that there are some coccolith proteins among the remaining 42 proteins in the dataset, hence our reference to these as 'an extended pool of candidates' (Supplementary figure 8).

We have replaced text from line 238 with the following to clarify:

We currently lack the computational tools to predict non-canonical secretory targeting pathways, and our knowledge of general cell biology in the haptophytes is furthermore extremely limited, meaning that it is entirely possible that some of the remaining 42 proteins are targeted to the CV. With this in mind, we consider these proteins as an expanded pool of candidates (Supplementary Fig. 8) and conservatively restrict COPROs to candidates with signal sequences.

Line 275. While the authors provide evidence that they successfully isolated protococcoliths bound by CV membrane, it is important to acknowledge the close association of ER and reticular body membranes that have been demonstrated through numerous TEM ultrastructure studies. Thus, the surrounding membrane(s) of isolated CVs are not likely comprised of just a single lipid vesicle membrane. There will be numerous 'extra-vesicular' materials. Indeed, the authors themselves find evidence of actin and myosin in these samples (see also line 385), highlighting the tight association of the CV membrane with the extra-vesicular environment, including adjacent ER-like membranes. Could the V-type ATPase signal in fact be more associated with the adjacent ER and not the CV itself?

We agree with the reviewer that not only CV membrane could be associated with the isolated proto-coccoliths, but also membrane fragments of the reticular body and possibly the ER. We have changed the text as follows to make the reader aware of this:

Altogether, these data suggest that the enriched proto-coccoliths may be enclosed in a membrane, which for simplicity we refer to as CV membrane here. In *Emiliana*, however, the CV membrane includes not only the membrane in contact with the calcite crystals, but also that of the reticular body, an organelle closely associated and partly fused with the CV⁴⁵.

We cannot exclude the possibility that ER membrane is present in our proto-coccolith preparations and that the detected V-type ATPase is located in this membrane. To make the reader aware of this, we have added the following sentence:

However, since no data are yet available on the *in vivo* localization data of this enzyme in *Emiliana*, the presence of ER membrane, which is closely associated with the CV⁴⁵, could also explain its presence in our proto-coccolith preparations

Related to the above remarks, interpretation of the SEM done to visualize the CV membrane of isolated protococcoliths (Supplementary Figure 9) needs to be revised or excluded. The authors interpret the different contrast of backscattered electron images (Panel B) to indicate CV membrane (low signal/darker greyscale) compared to the calcite (higher signal/brighter greyscale). The areas interpreted as membrane are almost certainly the organic baseplate

scale and not the very thin lipid bilayer of the vesicle if it was present. The escape volume of BSE is much deeper than SE and unlikely to give sufficient signal/contrast from just the CV membrane, if present, using the column conditions stated. Also classical freeze fracture typically allows for low energy secondary electron imaging from the cleaved lipid membrane surfaces. I do not see typical evidence one could expect indicating the presence of a membrane in the panel A image. The supplementary figure 9 is not sufficiently compelling in my view. Cryo-Electron tomography with TEM would be the best approach to resolve the architecture of the membranes and associated material of isolated protococcoliths.

We agree with the reviewer that freeze fracture SEM is not the most compelling EM technique for visualizing membranes. However, we would like to argue that the conclusion we drew, namely that organic material is enclosing the calcite (line 287), is still valid. We also like to point out that nowhere we did claim that this organic material is a membrane but only that it could be a membrane. Note that the SEM figure now is presented as Supplementary figure 11 in the new version of the manuscript.

1) '... the escape volume of BSE is much deeper than SE and unlikely to give sufficient signal/contrast from just the CV membrane, if present, using the column conditions stated.'

We agree that the escape volume of BSE is much deeper than that of SE. However, the thickness of the escape volume however depends on the imaging conditions. For image acquisition, we used a low accelerating voltage (1.2 kV) and JOEL's semi-in-column detector, with the r-filter set to BS mode. Under these conditions, only the BSE with energies close to the primary beam are detected (see the official documentation: <https://documents.uow.edu.au/content/groups/public/@web/@aiim/documents/doc/uow154681.pdf>).

To get an idea of the depth from which most of the BSE escape under our imaging conditions, we performed a Monte Carlo simulation on organic material with a density comparable with the one of lipid bilayers for 10^6 electrons using CASINO V3.

From the simulated trajectories, the maximum depth of each escaped backscattered electron was determined along with its energy. All the simulated electrons were then filtered for those that have an energy >1 kV, i.e. electrons whose energy is more than 80 % of the energy of the primary beam used for imaging. This revealed that 95 % of the BSE with >1 kV originated from a depth of about 5 nm, which corresponds to the thickness of a lipid bilayer.

To clarify the imaging conditions and the origin of the BSE, we have added the following to the legend of Supplementary Fig. 11:

Images were acquired with an acceleration voltage of 1.2 kV and the semi-in-column detector, with the r-filter set to BS mode. Monte Carlo simulations on organic material with a density comparable to lipid bilayers and for 10^6 electrons revealed that 95 % of the BSE with an energy of >1 kV, which were imaged, originated from a depth of about 5 nm.

2) 'The areas interpreted as membrane are almost certainly the organic baseplate scale and not the very thin lipid bilayer of the vesicle if it was present.'

While we cannot fully exclude in the central area of the coccoliths that the base plate contribute to the BSE signal, we can exclude this for areas outside of coccoliths such as the area marked by the orange arrow.

3) '... classical freeze fracture typically allows for low energy secondary electron imaging from the cleaved lipid membrane surfaces. I do not see typical evidence one could expect indicating the presence of a membrane in the panel A image.'

The samples were prepared by slush freezing. They got exposed at the surface of the ice before freeze fracturing. The coccoliths were found aligned in lines along the freezing fronts (probably because they were phase separating from the ice). We think that thermal shock and their movement associated with the freezing process caused them to brake. As a result, the resulting images differ somewhat from classical SEM freeze-fracture images.

Line 331 -334. The authors suggest their analysis of coccosphere proteins may be compromised by the production of extracellular vesicles. Please discuss how this dynamic exchange of plasma-membrane derived materials from cell to extracellular space affects your interpretation. These vesicles, which may have little to do with calcification per se could be the origin of several of the proteins considered to be calcification-specific.

This experiment was not designed to unambiguously identify calcification-related proteins on its own. Since we know that the internal contents of the CV are delivered to the coccosphere (i.e. coccoliths and CAP and associated molecules), we expect the coccosphere to be enriched in calcification-related proteins. However, we do expect proteins unrelated to calcification also to be present in the coccosphere, for example proteins involved in nutrient acquisition or biotic interactions. This means that we only consider presence in the coccosphere as a strong indication of involvement in calcification in combination with other evidence. Thus, our interpretation is not affected by the presence of membrane proteins, putatively derived from extracellular vesicles in the coccosphere. However, the observation does raise the interesting question of how CV production is associated with calcification, a topic that has not been explored in any depth.

To clarify our position on the interpretation of the coccosphere experiment we have made the following alterations to the text:

Line 333:

For this reason, we decided to examine the proteome of the coccosphere (dataset 4). It is important to note that we also expect proteins unrelated to calcification, such as proteins involved in nutrient uptake or biotic interactions to be present in the coccosphere, so orthogonal lines of evidence are crucial to the interpretation of this experiment.

333 possibly originating from the plasma membrane.

Line 392- This is a significant finding

Line 380. This is very intriguing!

Line 416. I found the supplementary note 7 tables confusing/unclear to read. Did the authors actually do PCR using the Mackinder et al. primer sets on each of these samples, or are all the ticks and crosses simply indicating in-silico predictions of whether the candidate genes would be amplified? Some additional explanation and guidance on how to read the tables would be helpful.

We thank the review for pointing out the need for clarification. These tables are indeed based on in silico predictions using the MacKinder et al primer sequences. We have updated the text to emphasise this.

The introductory sentence of Supplementary Note 7 has been updated to:

We used in silico searches with the primer sequences detailed by MacKinder et al. 2011 to see if we predict amplification of the same genes from the Emihu2 gene models as were targeted by MacKinder et al. This analysis makes it clear which qPCR results from the MacKinder et al work can be compared with our data, and which cannot.

Lines 424-433 this section is too speculative. For example, how could the SLC24 transporter be involved in regulating Ca²⁺ in the coccosphere (extracellular) microenvironment when seawater Ca²⁺ is ~ 10 mM? Please clarify the rationale here. Another example; if V-type ATPase proteins are up-regulated in N-cells vs C-cells then, using previous criteria, this should equally be interpreted evidence against a role in calcification correct? Also, because production of calcite from bicarbonate in the CV produces H⁺, over-acidification of the CV will happen when H⁺ removal is compromised, not necessarily because of over activation of a V-type ATPase.

We can envision several scenarios for how the SLC24 transporter can play a role in the calcium pathway for calcification. However, all scenarios are purely speculative, and may confuse the reader. Therefore, we decided to remove the discussion of the SLC24 transporter from the manuscript.

V-type ATPase: We prefer to avoid the reasoning that upregulation in N-cells vs. C-cells is evidence against a role in calcification. This is because factors involved in the regulation of cellular processes can act in either a positive or a negative fashion and genetic screens to identify genes involved in cellular processes routinely turn up both positive and negative regulators. Thus, as stated in lines 377 – 380, we consider differential expression between N and C-cells to constitute evidence of a possible role in calcification, but purposefully do not attach particular significance to the direction of change.

We accept the reviewer's analysis that over-acidification of the CV could in theory equally be caused by a problem with H⁺ removal or by over expression of the V-type ATPase. However, our data only provides evidence of the latter, which is why we present it as a possible explanation for the phenotype of N-cells.

Line 455- A helpful advance/resolution of a long-standing question

Thank you.

Line 466 -End of sentence does not make grammatical sense. Please revise.

Modified. The revised sentence reads:

This provides further reassurance that a high proportion of the proteins we identified are possibly related to calcification.

Line 594 Italics needed for species

Modified.

REVIEWERS' COMMENTS

Reviewer #1 (Remarks to the Author):

In this revised version of their manuscript, the authors have made more than substantial changes, in comparison to version 1. I must underline that they have entirely and precisely responded to the points I was asking clarification for (on both form and content). They have also lightened their figures (as requested) and reshaped their final concluding figure, now figure 8, which provides a solid conceptual framework to work with. Consequently, I do not see any reason not to accept the publication of this very informative and well-designed paper.

Reviewer #2 (Remarks to the Author):

The authors have responded to my remarks satisfactorily and I have no further comments other than the supplementary 7 notes showing the amplification predictions for previously used primers of calcification related genes. The explanatory text included in the revised version is helpful in confirming these are in-silico predictions only, however the tables are still ambiguous to read.

1. For example, the note and table 7.1 states:

"Calcium transporter genes that would be amplified by the Mackinder et al. CAX primers, and those found in the present proteomics study." Yet that table (column 1) includes primers for CAX and ECA. I suggest rewording 'primers for Ca²⁺ transporter genes'.

2. How are the ticks and crosses to be interpreted? For example first three rows reading left to right the CAX primer pair 4 from Mackinder is predicted to amplify this gene (tick in the Mackinder column), but would not amplify the predicted gene in the proteome from coccosphere, down day, Up-C, and down C cells (crosses) found in the authors' study? Or that the gene models are sufficiently different that the primers would not work even though the gene and protein are expressed in the current study? It's just not clear what an 'X' really means!

3. Should the first column be labelled "primer pairs from Mackinder et al."

4. The next three rows, which Na-Ca-K primers are being used here to test, are they from Mackinder (I assume so)?

5. Final 3 rows related to a Ca-ATPase and not a CAX, see comment above.

Please clarify these tables so the reader can easily interpret them and the significance, thank you.

RESPONSE TO REVIEWERS' COMMENTS

Reviewer #1 (Remarks to the Author):

In this revised version of their manuscript, the authors have made more than substantial changes, in comparison to version 1. I must underline that they have entirely and precisely responded to the points I was asking clarification for (on both form and content). They have also lightened their figures (as requested) and reshaped their final concluding figure, now figure 8, which provides a solid conceptual framework to work with. Consequently, I do not see any reason not to accept the publication of this very informative and well-designed paper.

Thank you.

Reviewer #2 (Remarks to the Author):

The authors have responded to my remarks satisfactorily and I have no further comments other than the supplementary 7 notes showing the amplification predictions for previously used primers of calcification related genes. The explanatory text included in the revised version is helpful in confirming these are in-silico predictions only, however the tables are still ambiguous to read.

1. For example, the note and table 7.1 states: "Calcium transporter genes that would be amplified by the MacKinder et al. CAX primers, and those found in the present proteomics study." Yet that table (column 1) includes primers for CAX and ECA. I suggest rewording 'primers for Ca²⁺ transporter genes'.

2. How are the ticks and crosses to be interpreted? For example first three rows reading left to right the CAX primer pair 4 from Mackinder is predicted to amplify this gene (tick in the Mackinder column), but would not amplify the predicted gene in the proteome from coccosphere, down day, Up-C, and down C cells (crosses) found in the authors' study? Or that the gene models are sufficiently different that the primers would not work even though the gene and protein are expressed in the current study? It's just not clear what an 'X' really means!3. Should the first column be labelled "primer pairs from Mackinder et al." 4. The next three rows, which Na-Ca-K primers are being used here to test, are they from Mackinder (I assume so)?5. Final 3 rows related to a Ca-ATPase and not a CAX, see comment above.

Please clarify these tables so the reader can easily interpret them and the significance, thank you.

We like to thank the reviewer for the suggestion to clarify the information provided in Supplementary Note 7. We have revised the tables and table legends in Supplementary Note 7 as follows:

Supplementary Note 7-1: Calcium transporter genes from the Emihu2 genome which were identified in the present proteomics studies or that we predict would be amplified by primers detailed in Table 1 of MacKinder *et al.* (2011). If the gene is predicted to be amplified by the MacKinder primers, then the name of the primer is given, otherwise a cross indicates no amplification is predicted. In the former case it can be concluded that the q-RT PCR data from MacKinder *et al.* (2011) may be reporting on the expression of the Emihu2 gene in question. If the protein product was identified in one of the four proteomics datasets a tick is present in the appropriate column. The absence of the protein in the proteome data set is indicated by “-”. “Down in day” means down regulated in the light in the Light vs. Dark dataset. “Up in C-cells” means upregulated in C-cells in the C-cell vs. N-cell dataset. “Down in C-cells” means downregulated in that dataset.

Emihu2 transcript or protein	Transporter type	MacKinder 2011	Coccosphere	Down in light	Up in C-cells	Down in C-cells
EhG30522.1	CAX	CAX4 primer pair	-	-	-	-
EhG30522.2	CAX	CAX4 primer pair	-	-	-	-
EhG18640.1	CAX	CAX4 primer pair	-	-	-	-
EhG40126.1 (.15)	Na ⁺ /Ca ²⁺ +K ⁺ exchanger	x	✓	✓	-	✓
EhG2170.2	CAX	x	-	-	✓	-
EhG13953.3	Na ⁺ /Ca ²⁺ +K ⁺ exchanger	x	-	-	-	✓
EhG33932 (iso 1 - 3)	ECA2	(ECA2 primer pair)	-	-	-	-
EhG853 (iso 1 - 4)	ECA2	(ECA2 primer pair)	-	-	-	-
EhG20651.3	ECA2	x	-	-	-	✓

Supplementary Note 7-2: AEL1 genes that would be amplified by the MacKinder *et al.* (2011) “AEL1” primers from Table 1, and those found in the present proteomics study. If the gene is predicted to be amplified by the MacKinder primers, a tick is present in the MacKinder column and further details of the MacKinder results are provided. If the protein product was identified in one of the four proteomics datasets, a tick is present in the appropriate column. The absence of the protein in the proteome data set is indicated by “-”. “Down in day” means down regulated in the light in the Light vs. Dark dataset. “Up in C-cells” means upregulated in C-cells in the C-cell vs. N-cell dataset. “Down in C-cells” means downregulated in that dataset.

	MacKinder 2011	Coccosphere	Down in day	Up in C-cells	Down in C-cells
EhG4408 (EhG4408.2 identified in proteomics)	✓ All isoforms 1-7 would be amplified. 2N-specific; up in C-cells vs N-cells	-	✓	-	✓
EhG4885 (iso 1 - 3)	x	-	-	-	-
EhG31923.6	x	-	-	-	✓
EhG909.1	x	-	-	-	✓

EhG5929.10	x	✓	✓	-	✓
------------	---	---	---	---	---

Supplementary Note 7-3: V-type ATPase genes found in the Emihu2 genome and either identified in the present proteomics study or predicted to be amplified by the MacKinder *et al.* (2011) primers. If they are predicted to be amplified there is a tick in the MacKinder 2011 column. If they were identified in a proteomics data set there is a tick in the relevant column. The absence of the protein in the proteome data set is indicated by "-".